# SWE-RL: Advancing LLM Reasoning via Reinforcement Learning on Open Software Evolution

**Yuxiang Wei**[12]    **Olivier Duchenne**[1]    **Jade Copet**[1]    **Quentin Carbonneaux**[1]

**Lingming Zhang**[2]    **Daniel Fried**[13]    **Gabriel Synnaeve**[1]    **Rishabh Singh**[1]    **Sida I. Wang**[1]

[1]Meta AI    [2]University of Illinois Urbana-Champaign    [3]Carnegie Mellon University

✉ ywei40@illinois.edu   ✉ sida@meta.com

https://github.com/facebookresearch/swe-rl

## Abstract

The recent DeepSeek-R1 release has demonstrated the immense potential of reinforcement learning (RL) in enhancing the general reasoning capabilities of large language models (LLMs). While DeepSeek-R1 and other follow-up work primarily focus on applying RL to competitive coding and math problems, this paper introduces SWE-RL, the first approach to scale RL-based LLM reasoning for real-world software engineering. Leveraging a lightweight rule-based reward (e.g., the similarity score between ground-truth and LLM-generated solutions), SWE-RL enables LLMs to autonomously recover a developer's reasoning processes and solutions by learning from extensive open-source software evolution data—the record of entire software development cycles, including code snapshots, code changes, and events such as issues and pull requests. Trained on top of Llama 3, our resulting reasoning model, Llama3-SWE-RL-70B, achieves a **41.0%** solve rate on SWE-bench Verified—a human-verified collection of real-world GitHub issues. To our knowledge, this is the best performance reported for medium-sized (<100B) LLMs to date, even comparable to leading proprietary LLMs like GPT-4o. Surprisingly, despite performing RL solely on software evolution data, Llama3-SWE-RL has even emerged with generalized reasoning skills. For example, it shows improved results on five out-of-domain tasks, namely, function coding, library use, code reasoning, mathematics, and general language understanding, whereas a supervised-finetuning baseline even leads to performance degradation on average. Overall, SWE-RL opens up a new direction to improve the reasoning capabilities of LLMs through reinforcement learning on massive software engineering data.

## 1   Introduction

The application of large language models (LLMs) to software engineering (SE) tasks has received significant attention, with researchers exploring their potential to automate various complex SE tasks, such as library-level and complex code generation (Zhuo et al., 2024; Zhao et al., 2024), real-world bug/issue resolution (Xia and Zhang, 2022; Jimenez et al., 2023; Yang et al., 2024c), and software testing (Deng et al., 2023; Jain et al., 2024a). Among these tasks, SWE-bench (Jimenez et al., 2023)—a benchmark for solving real-world software issues—has emerged as a focal point of research efforts, and researchers have proposed various agentic (Yang et al., 2024b; Wang et al., 2024b; Gauthier, 2024) and pipeline-based (Xia et al., 2024; Örwall, 2024; Zhang et al., 2024) methods to push LLMs' real-world issue solving capability. However, most current techniques rely on powerful proprietary LLMs like GPT-4o (OpenAI, 2024a) or Claude-3.5-Sonnet (Anthropic, 2024a), where advancements are driven more by enhanced prompting strategies than by improvements in the underlying LLM.

39th Conference on Neural Information Processing Systems (NeurIPS 2025).

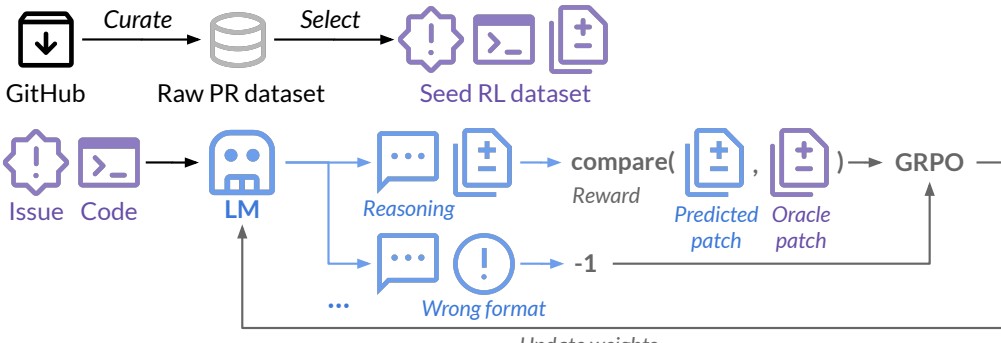

Figure 1: **Overview of SWE-RL.** We create a seed RL dataset from GitHub PRs, including issue descriptions, code context, and oracle patches. A policy LLM generates code edits via reasoning. Rewards are based on the similarity to oracle patches, with penalties for formatting errors. We optimize the policy using GRPO.

With the release of DeepSeek-R1 (DeepSeek-AI, 2025), reinforcement learning (RL) using rule-based rewards has become a crucial technique for enhancing the reasoning capabilities of LLMs across various downstream tasks, including coding (Zeng et al., 2025) and mathematics (Yeo et al., 2025). However, their effectiveness in SE tasks remains limited (DeepSeek-AI, 2025), and their substantial total parameter size (671B) poses challenges for researchers attempting to train them. For mathematics, the reward is generally defined as whether the answer predicted by the LLM can exactly match the ground truth (DeepSeek-AI, 2025). While in coding, existing RL research typically utilizes execution feedback (DeepSeek-AI, 2025; Gehring et al., 2025) as the reward signal and is limited to competitive programming tasks (Li et al., 2022; Jain et al., 2024b), where code is self-contained and easily executable. This is challenging to apply to real-world SE tasks due to the execution cost and lack of executable environments (Pan et al., 2024). Meanwhile, previous research (Ma et al., 2024; Pan et al., 2024; Xie et al., 2025) relied on proprietary teacher models and focused primarily on supervised fine-tuning (SFT), which, as we show in the paper, is less effective and less generalizable.

To address these limitations, we propose SWE-RL, the first RL method to improve LLMs on SE tasks by directly using rule-based rewards and software evolution data—the record of entire software lifecycle, including all code snapshots, changes, and events like PRs and issues. As shown in Figure 1, we begin by curating a comprehensive dataset of GitHub pull requests (PRs), which is then transformed into the seed dataset for RL. Each data item includes an issue, the corresponding code context, and the oracle patch merged by the PR. During RL, the policy LLM is tasked with solving a given issue through reasoning and producing the code changes. The code changes are then converted into a consistent patch format for reward calculation. If the response is incorrectly formatted, the reward will be $-1$; otherwise, the reward is a similarity score (between 0 and 1) of the predicted and the oracle patch calculated by Python's `difflib.SequenceMatcher` (Ratcliff and Metzener, 1988). Notably, we provide the complete content of each file in the input prompt, which implicitly teaches the model to reason about the precise fault locations before suggesting repair edits.

Applying SWE-RL to Llama-3.3-70B-Instruct (Dubey et al., 2024), our model Llama3-SWE-RL-70B solves **41.0%** of the issues in SWE-bench Verified (OpenAI, 2024), a human-verified subset of SWE-bench, with Agentless Mini, our pipeline-based scaffold built upon Agentless (Xia et al., 2024), featuring simplifications to match our RL process and enhancements for scaling. This performance is comparable to leading proprietary LLMs like GPT-4o (OpenAI, 2024a) and state-of-the-art among medium-sized LLMs with less than 100B total parameters. Our ablation studies demonstrate that Llama3-SWE-RL-70B significantly outperforms its Llama baseline. Additionally, we developed a competitive supervised fine-tuning (SFT) model from Llama-3.3-70B-Instruct using synthetic data generated in the Magicoder (Wei et al., 2024) style to enhance the chain-of-thought (Wei et al., 2022) process, employing the same seed as SWE-RL. We show that Llama3-SWE-RL-70B, trained with SWE-RL solely for solving issues, not only surpasses the SFT model in SWE-bench but also excels in other out-of-domain (OOD) tasks, including function-level coding (Chen et al., 2021; Liu et al., 2023), practical code generation with library use (Zhuo et al., 2024), code reasoning (Gu et al., 2024), mathematics (Hendrycks et al., 2021c), and general language understanding (Hendrycks et al., 2021b).

In these OOD tasks, Llama3-SWE-RL-70B even outperforms Llama-3.3-70B-Instruct, whereas the SFT model results in decreased performance.

In summary, our contributions are as follows:

- We introduce SWE-RL, the first RL approach specifically designed to enhance LLMs for SE tasks using software evolution data (e.g., PRs) and rule-based rewards.
- We develop Llama3-SWE-RL-70B, trained with SWE-RL on Llama-3.3-70B-Instruct. It achieves **41.0%** on SWE-bench Verified, the best performance among medium-sized language models (<100B) and even comparable to leading proprietary models like GPT-4o.
- We show for the first time that applying RL solely to real-world SE tasks, such as issue solving, can already enhance an LLM's general reasoning abilities, enabling it to improve on out-of-domain tasks like math, code generation, and general language understanding.

## 2 SWE-RL

To prepare the initial dataset for RL, we extract 273k high-quality PR seeds from the raw PR dataset we collected from GitHub. These seeds are selected based on specific heuristics. For example, a PR instance should include at least one linked issue, the issue should describe a bug-fixing request, and the code changes should involve programming files. For each seed, we extract the issue descriptions and code context, including all changed files and some relevant but unchanged files. These are converted to input prompts for the policy LLM. We also take the oracle patch from each seed, which will be used for reward calculation. More details are explained in Appendix A.

### 2.1 Reward modeling

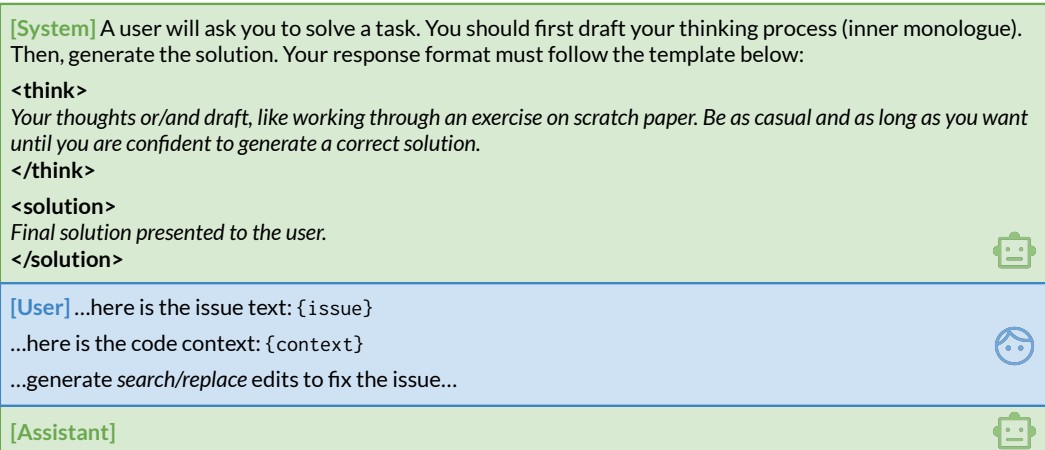

[System] A user will ask you to solve a task. You should first draft your thinking process (inner monologue). Then, generate the solution. Your response format must follow the template below:

**<think>**
*Your thoughts or/and draft, like working through an exercise on scratch paper. Be as casual and as long as you want until you are confident to generate a correct solution.*
**</think>**

**<solution>**
*Final solution presented to the user.*
**</solution>**

[User] ...here is the issue text: {issue}

...here is the code context: {context}

...generate *search/replace* edits to fix the issue...

[Assistant]

Figure 2: **Prompt template used to train Llama3-SWE-RL with SWE-RL.** Given an issue description and the corresponding code context, the policy LLM needs to generate search/replace edits (Xia et al., 2024) to fix this issue through reasoning. This is the only subtask we incorporate in the RL training. During inference, the LLM can generalize to tasks outside the training domain (e.g, file-level localization and test generation). For conciseness, we exclude certain prompt details, with the complete prompt template available in Appendix D.

We bootstrap the policy LLM with the prompt template shown in Figure 2. Assuming that the LLM has generated a rollout $o$ given an issue and its code context, the reward function is defined as follows:

$$\mathcal{R}(o) = \begin{cases} -1, & \text{if } o \text{ has wrong format,} \\ compare(\mathsf{patch_{pred}}, \mathsf{patch_{gt}}), & \text{otherwise.} \end{cases} \quad (1)$$

Here, $\mathsf{patch_{pred}}$ is the patch extracted from the LLM generation $o$ if it is correctly formatted, and $\mathsf{patch_{gt}}$ means the oracle patch for this issue.

In the implementation, we use Python's `difflib.SequenceMatcher` as the *compare* function, which returns a floating point between 0 and 1 indicating the sequence similarity, and adopt Group Relative Policy Optimization (GRPO) (Shao et al., 2024) for policy optimization.

Given a seed RL dataset $\mathcal{D}_{\text{seed}}$ where each item contains (1) issue, the issue description, (2) ctx, the code context required to solve this issue, including both files to repair and relevant files not intended for editing, and (3) $\text{patch}_{\text{gt}}$, which is the oracle patch. The input prompt for each data item is formed by instantiating the prompt template in Figure 2 with the issue description and code context, which we denote as $q = \text{form-prompt}(\text{issue}, \text{ctx})$. The policy LLM $\pi_\theta$ tries to solve the issue by generating code changes through reasoning, and produces multiple outputs $o_i$ for each input prompt $q$ given the group size $G$. Then, the policy LLM aims to maximize the following GRPO objective:

$$\mathcal{J}(\theta) = \mathbb{E}\left[\frac{1}{G}\sum_{i=1}^{G}\left(\min\left(\frac{\pi_\theta(o_i \mid q)}{\pi_{\theta_{\text{old}}}(o_i \mid q)}A_i, \text{clip}\left(\frac{\pi_\theta(o_i \mid q)}{\pi_{\theta_{\text{old}}}(o_i \mid q)}, 1-\epsilon, 1+\epsilon\right)A_i\right) - \beta D_{\text{KL}}(\pi_\theta \parallel \pi_{\text{ref}})\right)\right],$$

where $(\text{issue}, \text{ctx}, \text{patch}_{\text{gt}}) \sim \mathcal{D}_{\text{seed}}$, $q = \text{form-prompt}(\text{issue}, \text{ctx})$, and $\{o_i\}_{i=1}^{G} \sim \pi_{\theta_{\text{old}}}(\cdot \mid q)$. 

(2)

$\epsilon$ and $\beta$ are hyperparameters, and $\pi_{\theta_{\text{old}}}$ and $\pi_{\text{ref}}$ are the old and reference policy. The rewards $r_i = \mathcal{R}(o_i)$ and advantages $A_i = \frac{r_i - \text{mean}(r_1, \dots, r_G)}{\text{std}(r_1, \dots, r_G)}$ are calculated using the normalized rewards within each group following GRPO. $D_{\text{KL}}$ denotes the estimated KL-divergence (Schulman, 2020).

Our training approach conditions the model on the complete context of each file, implicitly forcing it to identify detailed fault locations before generating repair edits. This process inherently teaches the model both bug diagnosis and repair generation. However, during the evaluation on SWE-bench, Agentless Mini requires capabilities beyond generating repair edits, such as file-level fault localization, reproduction test generation, and regression test selection. Remarkably, even without explicit training on these subtasks, Llama3-SWE-RL can generalize to them through the RL process.

## 2.2 Aha moments and generalized reasoning capabilities

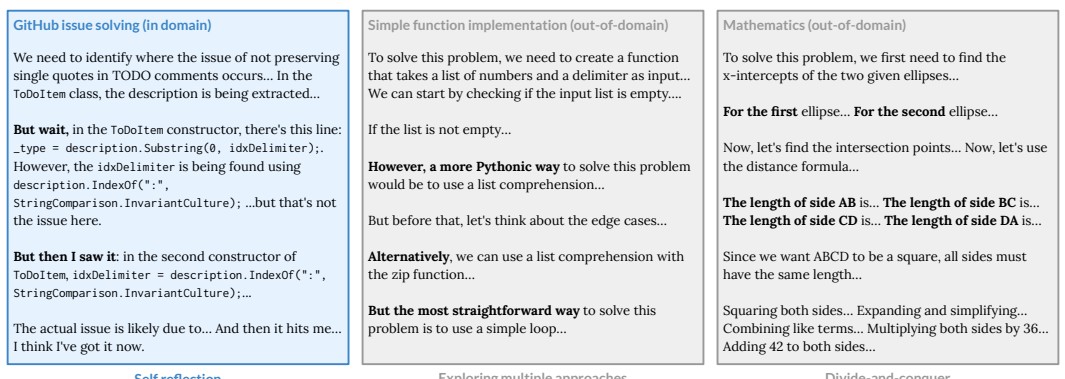

Figure 3: **Reasoning skills emerged from Llama3-SWE-RL-70B following the application of SWE-RL.** RL helps the model develop reasoning skills like self-reflection, exploring alternatives, and divide-and-conquer strategies, for both in-domain (e.g., issue-solving) and out-of-domain tasks (e.g., function implementation and mathematics).

**"Aha moments" on software engineering.** With the application of SWE-RL, we observe "aha moments" (DeepSeek-AI, 2025) where Llama3-SWE-RL exhibits emergent reasoning skills. To our knowledge, this is the first study demonstrating the existence of such aha moments in the realm of real-world software engineering tasks, confirming the findings of DeepSeek-R1 (DeepSeek-AI, 2025), which mainly focuses on competition coding and math. As shown in Figure 3, through RL, Llama3-SWE-RL-70B can allocate more thinking time to reflect on its initial assumptions during the issue-solving process. This behavior emerges naturally from the model's interaction with RL, rather than being explicitly programmed.

**General reasoning capabilities.** Surprisingly, we have identified additional aha moments where Llama3-SWE-RL acquires general reasoning abilities that are transferrable to various out-of-domain

tasks, such as function-level code generation and mathematics, although RL is applied exclusively to software issue solving. Figure 3 demonstrates that Llama3-SWE-RL is capable of reasoning through self-reflection, exploring alternative approaches, and solving complex problems by breaking them down into smaller subtasks. In §3.5, we demonstrate that Llama3-SWE-RL improves over even more out-of-domain tasks, including library use, code reasoning, and general language understanding.

# 3 Evaluation

## 3.1 Experimental setup

**Training configs.** Llama3-SWE-RL-70B is trained on top of Llama-3.3-70B-Instruct (Dubey et al., 2024) using SWE-RL for 1,600 steps with a 16k context window. We use a global batch size of 512, sampling 16 rollouts from each of the 32 problems in every batch. For every global step, a single optimization step is performed using Adam (Kingma and Ba, 2017). We train our models on 512 NVIDIA H100 GPUs; a training run takes approximately 32 wall-time hours.

**Scaffolding.** We have developed Agentless Mini on top of Agentless (Xia et al., 2024) as the underlying scaffold. Different from Agentless's multi-step localization, Agentless Mini focuses solely on file-level localization, delegating detailed reasoning to the repair step by providing the entire file contents in the input. This enables Llama3-SWE-RL to do more reasoning during SWE-RL and simplifies the RL process to focus on only one issue-solving task. Despite this simplification, Llama3-SWE-RL can still seamlessly complete other pipeline steps, benefiting from out-of-domain generalization through RL. Furthermore, while Agentless only employs one reproduction test per issue for reranking, Agentless Mini can use multiple reproduction tests, which has proven effective in our scaling analysis (§3.4). More details about Agentless Mini can be found in Appendix B.

**Evaluation setup.** We conduct evaluation on SWE-bench Verified (OpenAI, 2024), a subset of SWE-bench with 500 human-verified problems that can more reliably evaluate AI models' capability in solving real-world software issues. In the main evaluation (§3.2), we generate 500 patches for each problem using a 1.0 temperature, and use the top 30 reproduction tests for execution and reranking. Ultimately, only the highest-ranked patch for each issue will be submitted for SWE-bench evaluation to calculate the pass@1 score.

**SFT baseline.** To understand the advantages of SWE-RL, we also trained an SFT baseline, named Llama3-SWE-SFT-70B, for experiments in §3.3, §3.4, and §3.5. It is trained on top of Llama-3.3-70B-Instruct (Dubey et al., 2024) using a mixture of synthetic code editing data, Llama 3 (Dubey et al., 2024) coding SFT data, and Llama 3 general SFT data. The synthetic data is generated using an approach inspired by Magicoder (Wei et al., 2024), where high-quality PR data serves as the seeds for creating chain-of-thoughts and subsequent editing, as well as serving as the oracle for filtering. Llama-3.3-70B-Instruct is employed for this generation process. In contrast to our RL model, which only requires a seed PR dataset to trigger the RL loop, the SFT baseline needs synthetic data generation for chain-of-thoughts and additional data mix to ensure the dataset diversity and model generalizability. More details are explained in Appendix C.

## 3.2 Main results

Table 1 presents the pass@1 results on SWE-bench Verified for models that utilize open-source scaffolds, with models categorized by their size. From the table, we observe that Llama3-SWE-RL-70B achieves state-of-the-art results among small and medium-sized language models (<100B) by resolving 41.0% of the issues. Additionally, all other open-source baselines we compare, such as Lingma-SWE-GPT (Ma et al., 2024), SWE-Gym (Pan et al., 2024), and SWE-Fixer (Xie et al., 2025), include distilled outputs from GPT-4o or Claude-3.5-Sonnet in the training data. In contrast, Llama3-SWE-RL is trained solely with publicly available data through our reinforcement learning technique SWE-RL, without relying on any proprietary LLMs in the pipeline. Llama3-SWE-RL also sets a new record for Llama-based methods on SWE-bench. Additionally, we show that our RL-ed model is significantly better than the SFT baseline in this end-to-end setup, indicating that SWE-RL enhances the general reasoning ability of LLMs to solve issues.

Table 1: **Main results on SWE-bench Verified.** We include representative methods with open-source scaffolds. The scores are either collected from the SWE-bench Leaderboard (Jimenez et al., 2024) or from the corresponding reference.

| Model | Scaffold | SWE-bench Verified | Reference |
|---|---|---|---|
| | Model closed-source or size $\gg$ 100B | | |
| GPT-4o | SWE-agent | 23.2 | Yang et al. (2024b) |
| Claude-3.5-Sonnet | SWE-agent | 33.6 | Yang et al. (2024b) |
| GPT-4o | Agentless | 38.8 | Xia et al. (2024) |
| o1-preview | Agentless | 41.3 | OpenAI (2024b) |
| DeepSeek-V3[1] | Agentless | 42.0 | DeepSeek-AI (2024) |
| Claude-3.5-Sonnet | AutoCodeRover-v2.0 | 46.2 | Zhang et al. (2024) |
| Claude-3.5-Sonnet | Tools | 49.0 | Anthropic (2024b) |
| DeepSeek-R1[1] | Agentless | 49.2 | DeepSeek-AI (2025) |
| Claude-3.5-Sonnet | Agentless | 50.8 | Xia et al. (2024) |
| Claude-3.5-Sonnet | OpenHands | 53.0 | Wang et al. (2024b) |
| | Model size $\leq$ 100B | | |
| SWE-Llama-13B | RAG | 1.2 | Jimenez et al. (2023) |
| SWE-Llama-7B | RAG | 1.4 | Jimenez et al. (2023) |
| Lingma-SWE-GPT-7B | SWE-SynInfer | 18.2 | Ma et al. (2024) |
| Lingma-SWE-GPT-72B | SWE-SynInfer | 28.8 | Ma et al. (2024) |
| SWE-Gym-32B | OpenHands | 32.0 | Pan et al. (2024) |
| SWE-Fixer-72B | SWE-Fixer | 32.8 | Xie et al. (2025) |
| Llama3-SWE-SFT-70B | Agentless Mini | 36.2 | This paper |
| **Llama3-SWE-RL-70B** | **Agentless Mini** | **41.0** | **This paper** |

[1]Open-source Mixture-of-Experts model with 671B total and 37B active parameters

Table 2: Baseline comparison on SWE-bench Verified. In this experiment, we compare the repair-only performance of baseline LLMs by providing oracle localized files in the input context, without doing test generation and execution. We use greedy decoding by default, but for Llama-3.3-70B-Instruct, we include a 20-sample majority voting result at a temperature of 0.6 to improve formatting accuracy.

| Model | Setting | Correct format | Repair performance (oracle) |
|---|---|---|---|
| Llama-3.3-70B-Instruct | Greedy decoding | 12.2% | 5.4 |
| Llama-3.3-70B-Instruct | Majority voting | 44.6% | 16.6 |
| Llama3-SWE-SFT-70B | Greedy decoding | **96.2%** | 29.6 |
| **Llama3-SWE-RL-70B** | **Greedy decoding** | 95.6% | **34.8** |

### 3.3 Baseline comparison

To understand how much SWE-RL improves LLMs in solving sofware issues, we compare Llama3-SWE-RL with the corresponding Llama-3 and SFT baseline in Table 2, using Agentless Mini as the underlying scaffold. In this experiment, we also evaluate on SWE-bench Verified but focus on the models' repair ability. To achieve this, we provide oracle files in the context and let the model generate a single repair edit using greedy decoding, without incorporating additional pipeline steps such as localization and test generation. The table reveals that the base Llama-3.3 model struggles to produce correctly formatted code edits, even when using a 20-sample majority voting approach, where outputs with incorrect formats are pre-filtered. With SFT, most code edits generated by the language model are correctly formatted, and the repair performance shows significant improvement. However, Llama3-SWE-RL-70B demonstrates even greater enhancement in repair capabilities, although its format accuracy is slightly lower than that of the SFT version. This indicates that SWE-RL aids the LLM in better reasoning about issue solving and code editing.

## 3.4 Scaling analysis with more samples

Agentless Mini supports scaling both the number of repair samples and the number of generated reproduction tests. The difference in the number of samples may affect the reranking accuracy. In this section, we evaluate how the final pass@1 performance on SWE-bench Verified scales with the two factors.

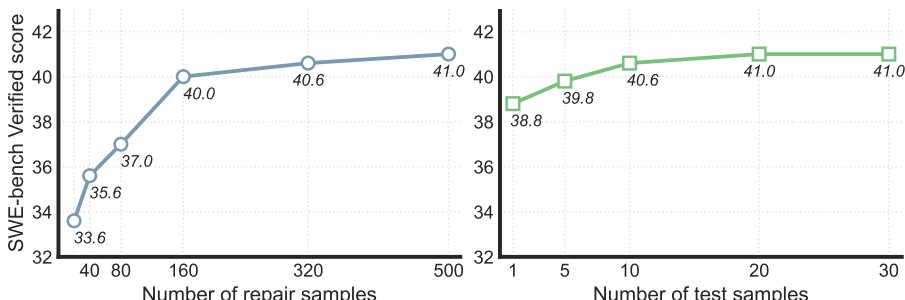

Figure 4: **Scaling analysis with more repair samples and more reproduction tests.** The figure on the left illustrates the resolve rate on SWE-bench Verified in relation to the number of repair samples, while maintaining a constant 30 test samples. Conversely, the figure on the right depicts the resolution rate as it varies with the number of reproduction test samples, with a fixed 500 repair samples.

Figure 4 shows that increasing both the number of repair samples and test samples enhances performance on SWE-bench. Notably, for repair samples, there is a significant score increase from 33.6 to 40.0 when the sample size is expanded from 20 to 160. However, beyond 160 samples, the improvement trend begins to plateau, with scores only rising slightly from 40.0 to 41.0 as the sample size increases to 320 and 500. Although the impact of adding more reproduction test samples is less obvious, there is still a gradual score improvement from 38.8 to 41.0 as the number of test samples increases up to 20. There is no difference between 20 and 30 test samples, suggesting a performance saturation point has been reached.

## 3.5 Generalizability of RL

Table 3: **Generalizability of Llama3-SWE-RL-70B beyond SWE-bench.** This table compares Llama-3.3-70B-Instruct, the SFT variant, and the RL model on five out-of-domain tasks, highlighting RL improvements and SFT declines. All experiments are done in a consistent setting using zero-shot greedy decoding. We report the macro average over category accuracy for MMLU and pass@1 for the others. In MATH, we use simple-evals's "Answer: ..." prompt format (OpenAI, 2024). However, only the RL model consistently follows the format requirements, so we also report MATH (lenient) to relax the constraint to include "\boxed...".

| Category
**Benchmark** | Llama-3.3-70B-Instruct | Llama3-SWE-SFT-70B | **Llama3-SWE-RL-70B** |
|---|---|---|---|
| Function coding
**HumanEval+** | 76.2 | 73.2 | **79.9** |
| Library use
**BigCodeBench-Hard (I)**
**BigCodeBench-Hard (C)** | **28.4**
**29.1** | 25.7
24.3 | **28.4**
**29.1** |
| Code reasoning
**CRUXEval-I**
**CRUXEval-O** | 60.5
61.9 | 68.4
75.1 | **71.6**
**75.5** |
| Math
**MATH (strict)**
**MATH (lenient)** | 63.2
70.9 | 54.0
71.7 | **73.7**
**73.7** |
| General
**MMLU** | 86.49 | 85.26 | **86.82** |

Llama3-SWE-RL is only trained with SWE-RL on issue-solving data. This raises a natural question whether such domain-specific training harms the performance on other tasks. To address this, we conduct an experiment in Table 3, evaluating the LLMs on five out-of-domain benchmarks, i.e., HumanEval+ (Chen et al., 2021; Liu et al., 2023) for function-level code generation, Big-CodeBench (Zhuo et al., 2024) for practical code generation with library use, CRUXEval (Gu et al., 2024) for code execution reasoning, MATH (Hendrycks et al., 2021c) for mathematical reasoning, and MMLU (Hendrycks et al., 2021b) for general language understanding. We also include the SFT baseline, which is finetuned on the same Llama-3.3-70B-Instruct model using issue-solving data, combined with general coding and dialog data.

From the table, it is evident that Llama3-SWE-RL-70B, trained with RL, outperforms both its base model and the SFT baseline. There are notable improvements in CRUXEval and MATH, where significant reasoning efforts are required to arrive at the final answer. Through SWE-RL, the model enhances its reasoning skills and dedicates more thinking effort to solving problems compared to other baselines. Although trained on a single task, SWE-RL enables the model to generalize its reasoning capabilities across various domains. In contrast, the SFT version, on average, underperforms relative to the original model.

Overall, our results suggest for the first time that reinforcement learning on real-world software data like PRs enables the model to acquire generalized reasoning skills, whereas supervised finetuning steers the language model towards a specific task distribution, leading to performance declines on tasks with lower emphasis, even when a meticulously curated data mix is used.

**Statistical significance analysis.** While small absolute gains (1–2 percentage points) on HumanEval or CRUXEval are, by themselves, unlikely to be statistically significant, we evaluate across multiple benchmarks and aggregate evidence, observing improvements that consistently favor RL. With Eval Arena (Wang et al., 2024a), we indicate that improvements of $> 0.8$ percentage points on MMLU, 3 points on CRUXEval, and $> 3$ points on the full MATH dataset are significant on their own; taken together, these results achieve significance at the 0.05 level.

### 3.6 Reward ablation

According to Equation (1), the reward design of SWE-RL allows different instantiations of the *compare* function. Throughout the paper, we adopt the sequence similarity between the predicted and the oracle patch, which is a continuous value from 0 to 1. We denote this type of reward as continuous. It is natural to compare this continuous reward with a discrete reward, where the *compare* function outputs 1 if the predicted and oracle patches exactly match each other, and 0 otherwise. We trained a variant of Llama3-SWE-RL with the discrete reward function, using the same training setup as in the continuous reward case.

| Reward type | Correct format | Repair (oracle) |
|---|---|---|
| Discrete | 94.2% | 29.0 |
| **Continuous** | **95.6%** | **34.8** |

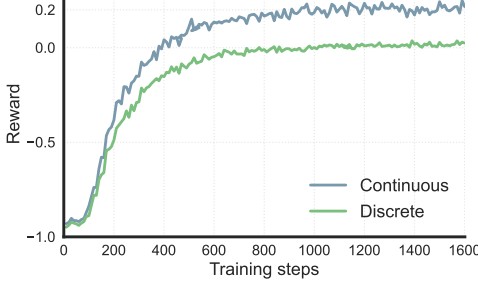

Figure 5: **Ablation on SWE-RL's reward functions and their training dynamics.** We compare SWE-RL using the default continuous reward function against a discrete reward. Repair (oracle) evaluates the repair-only performance using greedy decoding, with oracle files in the input context.

As shown in Figure 5, while the discrete and continuous reward functions lead to similar format accuracy, the continuous reward is more effective in enhancing the repair performance. From the training dynamics, we can see that discrete rewards grow slower than continuous rewards. Additionally, the average discrete reward remains approximately zero upon the completion of training, meaning it struggles to obtain patches exactly matching the oracles. This is because real-world patches are highly diverse and often cannot be easily matched. The continuous reward function

better captures partial correctness and incremental improvements, allowing the model to learn more nuanced and effective repair strategies.

# 4 Related work

## 4.1 Language models for software engineering

Large language models (LLMs), trained with billions to trillions of code tokens, have demonstrated outstanding performance in a wide range of coding tasks, including code generation (Chen et al., 2021; Li et al., 2023; Rozière et al., 2023; Guo et al., 2024; Wei et al., 2024; Lozhkov et al., 2024; DeepSeek-AI et al., 2024), code optimization (Cummins et al., 2024; Liu et al., 2024a), program repair (Xia and Zhang, 2022; Xia et al., 2023b; Jiang et al., 2023; Wei et al., 2023), and software testing (Xia et al., 2023a; Deng et al., 2023; Yuan et al., 2023; Schäfer et al., 2023; Lemieux et al., 2023). Initially, researchers primarily focused on single-shot code generation tasks, such as function-level (Chen et al., 2021; Austin et al., 2021; Hendrycks et al., 2021a; Li et al., 2022; Jain et al., 2024b), class-level (Du et al., 2023), and repository-level code completion (Ding et al., 2023; Liu et al., 2024b; Zhang et al., 2023). However, with the rapid development of LLMs, the performance on many popular single-shot code generation benchmarks like HumanEval (Chen et al., 2021), MBPP (Austin et al., 2021), and EvalPlus (Liu et al., 2023) has become saturated. Since the development of SWE-bench (Jimenez et al., 2023), which requires solving real-world GitHub issues, researchers start to work on improving LLMs' real-world issue-solving capability and have designed various scaffolds for SWE-bench. Two general types are (1) agentic scaffolds (Yang et al., 2024b; Wang et al., 2024b; Gauthier, 2024), where an LLM drives the decision-making process based on its past actions and observations through tool-based interaction with the environment; and (2) pipeline-based scaffolds (Xia et al., 2024; Örwall, 2024; Zhang et al., 2024), where an LLM goes through human-defined stages to solve a given issue. Generally, agentic methods are more general but require strong instruction-following and capable LLMs to drive the autonomous process, and can be computationally intensive due to multi-round interactions. In contrast, pipeline-based approaches are more specialized but efficient, with a focus on LLMs' pure code editing capability. Therefore, we designed our minimalist pipeline-based scaffold, Agentless Mini, to focus on the enhancements of Llama3-SWE-RL's core code editing capabiltiy.

## 4.2 Training software agents

While existing scaffolds have successfully leveraged proprietary language models to tackle real-world software engineering tasks, open models typically yield subpar results in these settings. Moreover, the most effective approach to enhancing real-world software engineering capabilities through training remains unclear. Recently, researchers have begun exploring the possibility of training open LLMs specifically for software engineering tasks, aiming to improve performance on benchmarks such as SWE-bench. For instance, Lingma-SWE-GPT (Ma et al., 2024) introduces 7B and 72B model variants that build on top of Qwen2.5-Coder-7B (Hui et al., 2024) and Qwen2.5-72B-Instruct (Yang et al., 2024a), using an iterative development-process-centric approach. SWE-Gym (Pan et al., 2024) presents the first open training environment for software engineering agents, significantly improving the performance of Qwen2.5-Coder's 7B and 32B variants on SWE-bench. More recently, SWE-Fixer (Xie et al., 2025) finetunes the Qwen2.5 base series, resulting in a 7B code retriever and a 72B code editor focused on efficient issue resolution, achieving notable best@1 improvements. Notably, all these works incorporate distilled samples from either GPT-4o or Claude-3.5-Sonnet in their training data and are built upon Qwen2.5 models. Their training objectives are all based on supervised finetuning. On the contrary, Llama3-SWE-RL is based on Llama 3 (Dubey et al., 2024) and trained through reinforcement learning (RL) using SWE-RL. The seed dataset for RL is sourced exclusively from publicly available repositories, allowing Llama3-SWE-RL to self-improve its issue-solving capabilities through the RL inscentive. Remarkably, Llama3-SWE-RL achieves the best performance among these models with a 41.0% solve rate on SWE-bench Verified (OpenAI, 2024), demonstrating for the first time that LLMs can already effectively address real-world issues through RL on real-world software artifacts.

# 5 Conclusion

We introduce SWE-RL, the first reinforcement learning (RL) approach to improve language models (LLMs) on software engineering tasks using software evolution data (e.g., PRs) and rule-based rewards. The resulting model, Llama3-SWE-RL-70B, achieves a **41.0%** solve rate on SWE-bench Verified, a human-verified collection of high-quality GitHub issues. This performance is state-of-the-art among medium-sized models and comparable to many proprietary LLMs such as GPT-4o. While SWE-RL is specifically applied to the issue-solving task, Llama3-SWE-RL has developed generalized reasoning skills through RL, demonstrating improved performance on out-of-domain tasks such as code reasoning, mathematics, and general language understanding. Overall, SWE-RL opens a new direction for enhancing the software engineering capabilities of LLMs through RL.

**Limitations.** Despite the promising results, our approach has several limitations. First, our reward implementation compares the sequence similarity between the predicted and oracle patch rather than their semantic equivalence. This may prevent the policy LLM from exploring alternative, functional equivalent solutions. Additionally, in Agentless Mini, the localization process is simplified to mapping repository structures to file paths, which lacks comprehensive context. Moreover, as a pipeline-based approach, Agentless Mini divides all steps into distinct inference stages. This "external structure" prevents the model from learning through interaction feedback and hinders its ability to consider the entire problem holistically.

## Acknowledgement

We thank Chris Waterson, Chris Cummins, and Volker Seeker for their assistance in debugging and testing Agentless Mini; Kunhao Zheng, David Zhang, and Taco Cohen, Jonas Gehring, Vegard Mella for their assistance in the online RL infra; Pierre Chambon, Mihir Sanjay Kale, Parth Thakkar, Michael Jiang, Sinong Wang, and Jingyue Shen for their involvement in the PR dataset discussion; Jannik Kossen for his helpful discussions in model training; Kamila Benzina, Rachel Kim, Megan Miller, Hannah Schubloom, and Madeline Hepler for their support in the review process; Steven Xia and Yinlin Deng for their aid in setting up Agentless; Jiawei Liu, Yifeng Ding, Terry Yue Zhuo, and Naman Jain for their valuable discussions; Albert Örwall for their help in resolving issues with Moatless EvalTools.

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

# A  Raw pull request data curation

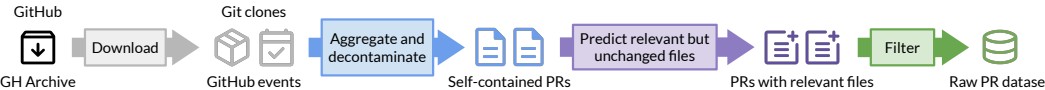

Figure 6: **Overview of SWE-RL's raw pull request data curation process.** The collected git clones and GitHub events are transformed into self-contained PR instances via decontamination, aggregation, relevant files prediction, and filtering.

Figure 6 provides a high-level overview of our process for curating the raw PR dataset for Llama3-SWE-RL. In the following paragraphs, we detail each step of the curation process. During data processing, we exclude all the repositories used by SWE-bench (Jimenez et al., 2023) to prevent data contamination.

**GitHub events and clones.** The goal of this stage is to recover all pull request details that human developers can inspect on GitHub. To achieve this, we need two sources of information: (1) all events that occur within a PR and (2) the source code of a repo before the changes introduced by the PR are merged. We derive all GitHub (GitHub, 2025) events from GHArchive (Grigorik, 2025), which contains all activity events data from GitHub. Our collection includes all GitHub events from Jan 1, 2015 to Aug 31, 2024.

To obtain source code, since pull requests often occur at different commit stages of a repository, we opt to use `git clone` to retrieve the entire repository with its commit history, rather than relying on the GitHub API (GitHub, 2022) to download specific code snapshots. Eventually, we successfully cloned and processed 4.6M repositories.

**PR data aggregation.** The collected events and git clones are disparate entities that require further processing before they can be used for training. At this stage, we focus on each PR individually and aggregate all pertinent information associated with it. This includes mentioned issues, user discussions, review comments, initial code contents, and subsequent commits and code changes.

First, we keep only merged PRs and gather all related conversational events for each PR, sorting them in chronological order. Next, using the `base_commit` and `head_commit` hashes of a PR, we retrieve the contents of all modified files indicated by its patch at the merge base of the two commits. The reason is that many PRs aim to merge back into the main branch, which may have undergone changes since the PR was created. By considering the merge base as the actual starting point for a developer working on the PR, we can more accurately understand the context of the changes. We save all intermediate commits and code changes made between the merge base and the head commit. Additionally, we extract the complete patch representing cumulative changes from start to finish. Finally, we scan each aggregated PR to identify patterns that resemble issues, and associate the matched issues with the corresponding PR. In the end, we have 24M aggregated PR instances.

**Relevant files prediction.** Currently, each pull request includes only the code files that have been modified. In our earlier experiments, we noticed that this approach let LLMs learn a bias: the model consistently generated edits for every code file presented and was unable to handle noisy files presented in the context. This issue was also mentioned in the finetuning experiment discussed in the SWE-bench paper (Jimenez et al., 2023). To mitigate this problem, we prompt Llama-3.1-70B-Instruct (Dubey et al., 2024) to generate a list of relevant but unmodified files given each PR description and the changed files, and include the contents of these files in our final dataset.

**Data filtering.** GitHub PRs can be quite noisy, so we implemented various filtering strategies to eliminate potentially harmful PRs. In designing these filtering rules, our goal is to maximize the recall of high-quality PRs while permitting a certain level of noise. First, we remove the bot-generated PRs whose title, description, or username contains keywords "[bot]", "dependabot", "renovate", "bump", or "automerge". Also, we remove PRs with empty changes or with extremely large number of changes (e.g., in some PRs, the developer uploaded a directory of data files by mistake). Additionally, we implemented a more fine-grained set of filters used in CodeLlama (Rozière et al., 2023) to examine each code change hunk. We then removed any PRs where these filters flagged all code changes. For example, this can exclude PRs with only lock file changes or version updates. Finally, this gives us around 11M unique PR instances. Before applying reinforcement learning, we select 273k high-quality PRs from these raw PRs, as discussed in §2.

# B    Agentless Mini

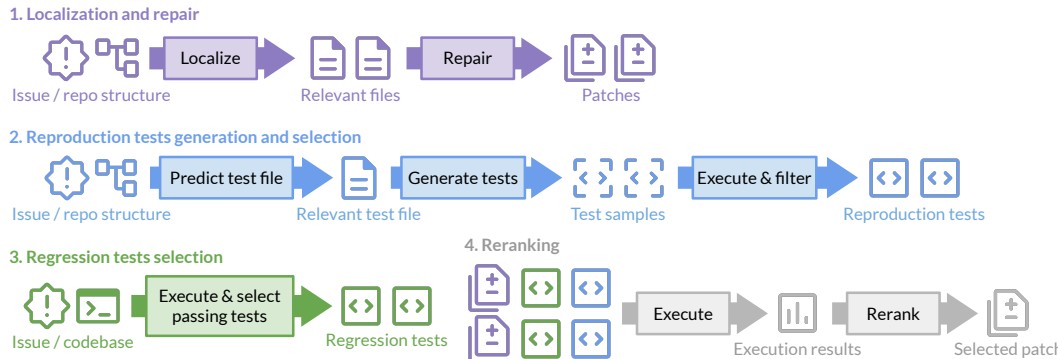

Figure 7: **The Agentless Mini scaffold.** The design emphasizes easy decomposition, parallelization, and scalability.

In addition to a model proficient in code editing, effectively tackling software engineering tasks, such as those found in SWE-bench (Jimenez et al., 2023), also requires a robust scaffold. Agentless (Xia et al., 2024) is one of the state-of-the-art scaffolds for SWE-bench at the time of writing. Building upon Agentless with various simplifications and enhancements, we developed Agentless Mini, a framework that prioritizes straightforward component decomposition, parallelization, and scalability. With Agentless Mini, each step's inference or execution compute can be independently scaled to enhance SWE-bench performance. In Figure 7, we present a detailed illustration of Agentless Mini's working principles. The following paragraphs will elaborate on each step and highlight the differences from Agentless.

**Localization and repair.** For localization, we employ a prompting-based approach that enables the model to predict relevant file paths based on a given issue and the repository's structure. Unlike Agentless, which involves two additional detailed steps to identify related elements and files, as well as a separate embedding model, Agentless Mini simplifies the process. It generates multiple samples of potentially problematic files from the model and consolidates them into unique sets for repair.

During the repair phase, the LLM is conditioned on the full content of the files to predict search/replace edits. We generate multiple repair samples from different location sets, ensuring a comprehensive exploration of the patch search space.

**Reproduction tests generation and selection.** Agentless samples reproduction tests for patch selection. Initially, multiple reproduction tests are generated based on an issue description, and one majority sample is selected after filtering. These tests must have distinct logic to output "Issue reproduced" and "Issue resolved" when the issue is reproduced or resolved, respectively. They are filtered based on whether they correctly output "Issue reproduced" when executed in the original codebase. Agentless Mini enhances this pipeline with two key improvements. First, instead of relying solely on the issue description, the model retrieves a relevant test file to guide test generation. Additionally, rather than selecting just one majority sample, Agentless Mini allows for the selection of multiple top test samples based on voting results. In our evaluation, using more test samples has proven beneficial for reranking (§3.4).

**Regression tests selection.** We select regression tests in the same manner as Agentless. Initially, we gather all passing tests for each issue by executing the code before any modifications. This step does not require model inference and needs to be performed only once. Subsequently, an additional, optional inference step is conducted to filter out passing tests that are supposed to fail after the issue is fixed. The rest of the tests will be marked as regression tests.

**Reranking.** Agentless Mini utilizes both regression and reproduction tests for reranking. For each issue, every applied patch is executed against the regression tests. The patches that result in the minimum number of existing test failures are selected. For each issue, we choose the top-$N$ reproduction tests and run each patch against these tests. If the execution outputs "Issue resolved", we mark the patch passing this test. We then adopt the dual execution agreement objective from CodeT (Chen et al., 2023). Specifically, patches $\mathcal{P}$ that pass the same set of reproduction tests $\mathcal{T}$ are denoted as a

consensus group. Each consensus group is scored using the formula $|\mathcal{P}| \times |\mathcal{T}|^2$. With this objective, consensus groups with patches passing more tests receive higher scores. Additionally, groups where more patches pass the tests are scored higher, although passing more tests is prioritized over having more patches. Finally, we identify the consensus group with the highest score and select the best patch from this group using majority voting.

## C  Synthesizing supervised-finetuning data

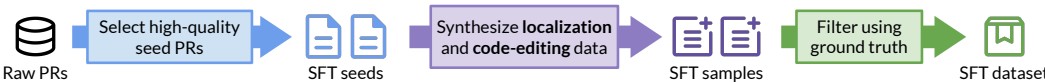

Figure 8: **Synthetic data pipeline for constructing SFT data.** We start by collecting high-quality seed PRs using heuristics, then generate synthetic localization and code-editing samples, and finally use the ground-truth edited files and patches to filter out incorrect samples.

Figure 8 shows our method of generating synthetic supervised-finetuning (SFT) data. The data generation pipeline is inspired by Magicoder (Wei et al., 2024), where the OSS-Instruct technique generates high-quality code instruction data from open-source seed snippets. We apply a similar methodology to generate both fault localization and code editing data using high-quality PR seeds.

**Collecting high-quality seed PRs.** To begin with, we extract high-quality PR seeds from the raw dataset we collected, as detailed in Appendix A. These seeds are chosen based on specific heuristics. For example, a PR instance should include at least one linked issue, the issue should describe a bug fix request, and the code changes should involve programming files.

**Localization and editing data synthesis.** We adopt Llama-3.3-70B-Instruct (Dubey et al., 2024) for data synthesis. For localization data, we prompt the model with the issue description, repository structure, and the paths of edited and relevant files as hints. We then ask the model to identify the relevant files for modification or review by generating a thought process, followed by a prioritized list of file paths. During filtering, we ensure that the model's response includes all files that are genuinely edited in the PR, and these files should be prioritized in the ranking.

Similarly, in terms of code editing, we synthesize code edits for a given issue, in search/replace format (Xia et al., 2024), by providing the ground-truth PR and patch as guidance to Llama-3.3-70B-Instruct. During the filtering process, we ensure that all search/replace blocks adhere to the correct format and that all search paths and blocks can be accurately matched within the input files' context.

**SFT baseline.** As discussed in §3.1, we meticulously constructed Llama3-SWE-SFT-70B as a strong SFT baseline. This SFT baseline is trained on a 16k context window for 2B tokens, where the training data consists of a mix of the aforementioned synthetic localization and editing data, as well as coding and general SFT datasets from Llama 3 (Dubey et al., 2024).

## D  Complete prompt

The following is a complete version of Figure 2:

```
PROMPT_TEMPLATE = """<|begin_of_text|><|start_header_id|>system<|
    end_header_id|>

A user will ask you to solve a task. You should first draft your thinking
    process (inner monologue). Then, generate the solution.

Your response format must follow the template below:
<think>
Your thoughts or/and draft, like working through an exercise on scratch
    paper. Be as casual and as long as you want until you are confident to
    generate a correct solution.
</think>
<solution>
```

Final solution presented to the user.
</solution><|eot_id|><|start_header_id|>user<|end_header_id|>

We are currently solving the following issue within our repository. Here is
    the issue text:
--- BEGIN ISSUE ---
{problem_statement}
--- END ISSUE ---

Below are some code segments, each from a relevant file. One or more of
    these files may contain bugs.

--- BEGIN FILE ---
```
{content}
```
--- END FILE ---

Please first localize the bug based on the issue statement, and then
    generate *SEARCH/REPLACE* edits to fix the issue.

Every *SEARCH/REPLACE* edit must use this format:
1. The file path
2. The start of search block: <<<<<<< SEARCH
3. A contiguous chunk of lines to search for in the existing source code
4. The dividing line: =======
5. The lines to replace into the source code
6. The end of the replace block: >>>>>>> REPLACE

Here is an example:

```python
### mathweb/flask/app.py
<<<<<<< SEARCH
from flask import Flask
=======
import math
from flask import Flask
>>>>>>> REPLACE
```

Please note that the *SEARCH/REPLACE* edit REQUIRES PROPER INDENTATION. If
    you would like to add the line '        print(x)', you must fully write
     that out, with all those spaces before the code!
Wrap each *SEARCH/REPLACE* edit in a code block as shown in the example
    above. If you have multiple *SEARCH/REPLACE* edits, use a separate code
     block for each one.<|eot_id|><|start_header_id|>assistant<|
    end_header_id|>

"""

---

