# OpenReview forum: "SWE-RL: Advancing LLM Reasoning via Reinforcement Learning on Open Software Evolution"
_NeurIPS.cc/2025/Conference — NeurIPS 2025 poster_

### Official Review · Reviewer_2vNV · 2025-06-24

**Clarity:** 4
**Significance:** 4
**Originality:** 3
**Rating:** 4
**Confidence:** 5

**Summary:**

This paper introduces a novel framework, SWE-RL, which uses reinforcement learning (RL) designed to enhance large language model (LLM) reasoning for software engineering (SE) tasks by leveraging software evolution data (GitHub pull requests). The authors fine-tune LLaMA-3-70B-Instruct using 'rule-based' rewards (patch similarity computed via Python library "difflib.SequenceMatcher"), with the resulting Model known as: __Llama3-SWE-RL-70B__. The proposed approach shows substantial performance by achieving a 41.0% solve rate on SWE-bench Verified, which outperforms medium-sized open models and rivals proprietary systems like GPT-4o. In addition, the derived model reveals emergent inference to out-of-domain (OOD) tasks (e.g., math, code reasoning) despite being trained exclusively on SE data.

**Questions:**

1. Have you validated whether the `difflib`-based reward encourages shallow formatting mimicry rather than semantic code transformations? I would recommend that the authors provide any adversarial examples or robustness tests.

2. What empirical evidence links SWE-RL training to improved general reasoning, beyond slightly higher pass\@1 scores on MATH and MMLU? Were alternative prompts or reasoning diagnostics used?

3. Why didn’t the authors include an LLaMA-3 SFT baseline using the same PR dataset to isolate the contribution of RL?

4. Can the authors report confidence intervals or run their experiments with different seeds to show variability and robustness?

5. What was the observed variance in the reward signal during RL training? Were instabilities observed?

**Ethical Concerns:**

["NO or VERY MINOR ethics concerns only"]

**Final Justification:**

The authors have meaningfully improved the clarity of the paper and addressed several of my main concerns, particularly by:
Clarifying the reward design. Authors have provided a rationale for using the difflib-based similarity metric, linking it to the plastic surgery hypothesis in program repair and emphasizing its scalability advantages over execution-based rewards. Additional experiments showed that combining similarity with execution-based reward outperforms execution alone.
Providing evidence for reasoning improvements. The authors supplied qualitative reasoning traces and quantitative measurements (longer “thinking length” and output length), suggesting that SWE-RL encourages multi-step reasoning behaviors.
Establishing stronger baselines. An SFT baseline trained on the same PR dataset was included, and end-to-end evaluations confirmed that RL yields consistent gains over SFT.
Statistical significance and stability. The rebuttal added statistical testing details (e.g., typical standard error on SWE-bench Verified, significance thresholds) and confirmed that training reward trends are stable with low variance, supported by concrete reward progression data.
Reproducibility commitments. The authors have pledged to open-source their work, and clarified hyperparameters to support replication.
These additions strengthen the technical soundness of the work and address many practical concerns regarding robustness and reproducibility.

However, some significant limitations remain:
Reward signal brittleness: While the authors argue for the efficiency of difflib, it still does not reason about control/data flow, side effects, or deeper semantic correctness, leaving potential for over-optimization on superficial similarity.
Limited ablation coverage: The lack of sensitivity studies for KL penalty, group size, and clipping remains an empirical gap. While the computational constraints are acknowledged, even a partial or theoretical exploration of these effects would strengthen the framework’s methodological grounding.
OOD generalization claims: Gains on MATH, HumanEval+, and CRUXEval are small; while the authors’ statistical arguments improve confidence, further targeted diagnostics (e.g., reasoning trace analysis on OOD tasks) would make the case more convincing.
Overall, SWE-RL is a timely and relevant contribution to the intersection of RL and LLMs for software engineering.

The authors have adequately improved my confidence in the reported results. However, I strongly suggest future work to address enhancing semantic depth in the reward function and expanding robustness analyses.

That said, I maintain a borderline accept rating, leaning toward acceptance given the strengthened empirical support, clearer framing, and practical impact potential.

**Limitations:**

The authors partially acknowledge limitations (e.g., lack of statistical testing), but omit critical discussion on the brittleness of their reward signal, the risk of overfitting to diffs, and the unexplained generalization claim. More transparency about the model’s weaknesses in semantic reasoning, reasoning trace depth, or vulnerability to reward hacking would strengthen the work.

**Quality:**

4

**Strengths And Weaknesses:**

This manuscript presents a rich set of valuable results that could potentially enhance the practice of SWE research and practice. However, some drawbacks hinder the validity of these contributions, which I believe the authors must address.

 __Strengths__

 On the one hand, this paper:

- Addresses a timely and under-explored area in LLM fine-tuning for code-based tasks that enhances general reasoning through SWE tasks. Specifically on static metrics like PR.

- Presents an actual use of PR-based 'diff' similarity as a reward signal for RL training on real-world code edits.

- Illustrates competitive performance on SWE-bench Verified (41.0%), which suggests that the fine-tuned model improves at long-context issue resolution.

- Claims generalization to OOD domains (MATH, HumanEval, MMLU), even though the model was trained only on SWE (PR-related) tasks.

-- Reuses available (open) tools and datasets (e.g., SWE-bench), paving the way for future research and reproducibility in principle.



 __Weaknesses__

On the other hand, four major areas of weakness pose serious issues for the authors to address:

1 **Quality:**

1.1. The reward function represented in Eq. (1), Pg. 3 is a bit shallow, based on the fact that it relied mainly on the string-level similarity (`difflib`), thus, prone to exploitation via formatting tricks or lexical mimicking, rather than genuine semantic improvements.

1.2. The policy LLM represented in Eq. (2), Pg. 4, interestingly, uses a KL Penalty for Regularization that penalizes the updated policy from drifting too far from the reference model, in this case, vanilla LLaMA-3, which preserves general capabilities while specializing in the SWE fine-tuning tasks. However, the KL Divergence Target Is Not Fully Defined: 𝜋_ref is introduced, but we do not know if it's frozen, periodically updated, or fixed to 𝜋_old

1.3. Group Size 𝐺 Sensitivity specifies the number of rollouts per prompt, which is quite critical but under-specified by the authors.

1.4. I would suggest that the authors include an ablation study on group size (𝐺) clipping (𝜖), and KL penalty 𝛽. Then, report the learning stability over time, particularly for low-reward categories.

1.5. No ablation is provided comparing SWE-RL to supervised finetuning (SFT) on the same PR data, leaving it unclear whether RL contributes meaningfully. The authors should be intentional with their empirical setup upfront to guide a clear understanding of the reported results.

1.6. The size or quality of the PR task is not considered as would be in a typical SWE setting. It's not clear to me if similar results would be observed with change request data, since PR is typical of the GitHub-related tool. Besides, on Pg. 5, the authors reported 500  patches and a temperature of 1.0. Why 500? Motivate this number and justify when the temperature was set to 1.0

1.7. The generalization claim to OOD domains is not adequately supported: improvements are marginal (1–2%) and lack statistical testing or multiple runs.

1.8. In Section 3, the authors seem to have omitted statistical testing (significance), as the authors claim, due to greedy decoding, which is incorrect. Variance still exists in task difficulty, decoding determinism, and sampling.

1.9. The SWE-bench results (Table 1) are overinterpreted: outperforming DeepSeekCoder-33B and WizardCoder-15B is expected given the model’s size and data alignment. However, claiming “comparable to GPT-4o” without GPT-4o’s SWE-bench Verified number is speculative. The authors should turn down their tune on reported observations.

1.10. Surprisingly, no analysis of reward variance, RL stability, or convergence is reported, despite the complexity of RL optimization.

2 **Clarity.**

__The authors should pay special attention to how scientific papers are written, specifically in words and the choice of color.__
2.1. Footnote formatting is inconsistent; footnotes in Table 1 clutter interpretation.

2.2. Equations and notation (e.g., Reward function R(o)) are inconsistently rendered.

2.3. Typos such as “sofware” in Section 3.3 suggest rushed proofreading.

2.4. Key terms like “aha moments” should be discouraged. They are used loosely and should be replaced with formal terminology.

2.5 The authors should properly define and cite rule-based rewards on first mention, before using them.

2.6. I was expecting to see more SWE papers in the Introduction when announcing SE tasks/bugs/Issues, and software testing.

2.7. In Figure 1. IS there any dependency between these disconnected processes?
From GitHub to Seed RL dataset and Issues/Code to GRPO. Can the authors explain how these two are related diagrammatically?

2.8. ‘SWE tasks‘ is ambiguously defined in this paper. Can the authors properly define an SWE task, such as PR? To elucidate the process and explain why PR was selected for this study and not other SWE tasks? Besides, PR is typical of GitHub technology. What other/similar processes exist with other workflows?

2.9. Pg.2 Line 48-50: "we propose SWE-RL, the first RL method to improve LLMs on SE tasks by directly using rule-based rewards and software evolution data—the record of entire software lifecycle, including all code snapshots, changes, and events like PRs and issues." This statement is too assuming. I would advise the authors to familiarize themselves with SWE conferences on KLLM4Code and AIWare papers

2.10. Line 50: Producing the code change, how? Explain what is code change is and how the LLM produces it at this stage

2.11. Line 55-57: Add a visual to support this description, showing files with code change set patch sets. What similarity measures did you use? Cosine?
Lines 58-59: Given that fault localization is a difficult problem in SWE, the author should explain how this process was done.

What does the Python ‘difflib’ do?

3 **Significance:**

3.1. This work aims to advance SWE domain expertise by developing models fine-tuned or trained with SWE-specific data, which could have a broader impact. Still, the lack of deeper reward design and validation makes the current framework fragile.

3.2 If general reasoning improvement from domain-specialized RL could be proven, the implication would be important. However, this remains unsubstantiated. Contrary to this, how could we be certain that a vanilla LLM would not generalize to OOD tasks as such? I would suggest that the authors experiment in a control environment before concluding on OOD

4 **Originality:**

4.1. SWE-RL appears novel in combining GitHub PR diffs and RL, but builds directly on SWE-bench and DeepSeek-style setups. The reward shaping and update rule (GRPO) is inherited without critical innovation.



-------
I would suggest that the authors reference SWE papers from among these top-tier venues: ICSE/ICSME/MSR/SANER + TSE/TOSEM/EMSE/JSS Papers

---

> ### Author Rebuttal · Authors · 2025-07-31
>
> Dear Reviewer 2vNV, we deeply appreciate your insightful feedback and suggestions for our work. In our responses below, we address each primary question (denoted as Q) and comment (denoted as C). Should there be any misunderstandings of the questions, please kindly let us know; we are eager to communicate with you throughout the discussion period.
>
> > Q1: …difflib-based reward encourages shallow formatting mimicry rather than semantic…
>
> We want to kindly highlight that the main novelty of SWE-RL lies in the insight that real-world software bugs often follow contextualized patterns, aligning with the plastic surgery hypothesis [1] in program repair. The difflib-based reward can efficiently capture partial correctness of code changes and makes large-scale training possible; execution-based rewards require expensive environment setup and heavy data curation [2], limiting scalability.
>
> As a result, SWE-RL requires non-trivial reasoning to localize errors and generate patches correctly to solve the software issues. During RL, such reasoning patterns emerge, involving planning and backtracking. We show one example below. Please also kindly refer to Figure 3 in the paper for more examples.
> ```
> We need to identify where the issue of not preserving…
> But wait, in the ToDoItem constructor, there's this line: _type = description…
> but that's not the issue…
> But then I saw it: in the second constructor…
> The actual issue is likely…
> And then it hits me…
> I think I've got it…
> ```
>
> Meanwhile, test-based reward is never perfect. Sometimes, it can be worse than similarity metrics where incorrect patches pass all the tests but miss the true intention. For example, [3] shows that the insufficient tests in SWE-bench impacts 40.9% of SWE-Bench Lite and 24.4% of SWE-Bench Verified leaderboard entries.
>
> In the table below, we also did an additional experiment showing that combining SWE-RL reward with execution is superior to applying execution-only reward alone. Please kindly refer to Q1 from Reviewer n4Dr for more details.
>
> |Setting|Pass@1|
> |-|-|
> |Baseline|0.8%|
> |Execution-only|11.0%|
> |Execution + SWE-RL reward|**14.2%**|
>
>
> > Q2: What empirical evidence links SWE-RL training to improved general reasoning…
>
> Great question. In the training, the model needs to generate a correct patch in one step conditioned on the relevant code context that is typically long. To maximize the reward, it performs extensive and accurate reasoning to find the correct edit location and to produce a correct patch. This process helps the model bootstrap its general reasoning capabilities that are transferable to other tasks. In the table below, we can observe that the maximum thinking length (measured in character count) continuously increases during training:
>
> |Training steps|Max thinking length|
> |-|-|
> |400|**15546**|
> |800|**17704**|
> |1600|**20203**|
>
> Similarly, in the following table, we measured the average output length of the original, SFT, and SWE-RL checkpoints on HumanEval+ and MATH. After RL, the reasoning is longer, supporting our assumption that the model acquires general reasoning ability:
>
> |Benchmark \ Setting|Baseline|SFT|RL|
> |-|-|-|-|
> |HumanEval+|746|657|**1622**|
> |MATH|1876|1643|**2552**|
>
> Also, we can observe qualitative examples like the one shared in Q1 and those in Figure 3, indicating that SWE-RL incentivizes the model to develop new reasoning strategies.
>
> We use the most appropriate prompts during evaluation and did an additional experiments to align the prompts. As shown below, RL performs the best regardless of the prompt change. Please kindly refer to Q2 from Reviewer ZGKJ for more details.
>
> |Benchmark|SFT (prompt in paper)|SFT (RL prompt)|RL (RL prompt)|
> |-|-|-|-|
> |HumanEval+|73.2|70.1|**79.9**|
> |MATH|71.7|70.6|**73.7**|
>
> > Q3: Why didn’t the authors include an LLaMA-3 SFT baseline…
>
> Please kindly note that we already included the SFT baseline in Table 2 for repair performance evaluation. The SFT checkpoint uses the same PR dataset as in RL. We also did an additional end to end evaluation below, showing that RL is superior to SFT.
>
> |Setting|Pass@1|
> |-|-|
> |SFT|36.2%|
> |RL|**41.0%**|
>
>
> > Q4: Can the authors report confidence intervals…
>
> Absolutely. For SWE-bench Verified, the typical paired standard error std(A-B) is 2% for models A and B. Thus two models typically need to differ by 1.96 * 2% (~4%) to be significantly different at the 0.05 level. Coincidentally, std(A) is also around 2% for SWE-bench Verified.
>
> > Q5: What was the observed variance in the reward signal…
>
> Good point. As shown in Figure 5, the training reward steadily increased during training. It’s overall stable and the variance is low.
>
> > C1: …the KL Divergence Target Is Not Fully Defined…
>
> During RL, $\pi_{\rm{ref}}$ is always set to be the original Llama 3 and kept frozen. Our objective formula and KL definition follows the standard of existing work [4]. We will clarify this in the revision.
>
> > C2: Group Size… under-specified by the authors.
>
> Please kindly refer to Section 3.1 of our paper. We described that we sampled "16 rollouts from each of the 32 problems in every batch", so the group size is 16. We will make it more clear in the revision.
>
> > C3: …include an ablation study on group size, clipping, and KL penalty…
>
> Thanks for the suggestion. Our hyperparameter choices largely follow the existing best practice in the literature [4]. Unfortunately, we do not have the compute budget to ablate all combinations of the hyperparameters, but this is a valuable suggestion and we will include more ablations in the future when we have sufficient computing resources.
>
> > C4: The size or quality of the PR task is not considered… Why 500…
>
> We did rigorous filtering to ensure the quality of the PRs used in the RL training. Please kindly refer to Appendix A for more details regarding the data curation process. Regarding the 500 patches, we demonstrated in Figure 4 in the paper that 500 patches enabled continuous test time scaling of the issue solve rate on SWE-bench Verified.
>
> > C5: …claim to OOD domains is not adequately supported…
>
> 1-2% performance gains on HumanEval or CRUXEval are not likely to be significant by themselves (at the 0.05 level). That is why we reported on multiple evaluations where all results are consistently in favor of RL, thus giving increased significance (for example, via Fisher’s combined probability test). To support this, we conducted additional statistical testing (link omitted due to rebuttal policy), showing that >0.8% on MMLU (14k examples), 3% on CRUXEval, and >3% on the full MATH are already significant. So the combination of our results will reach significance at the 0.05 level. We will include this statistical analysis in the revision of our paper.
>
> > C6: …Variance still exists…
>
> Indeed greedy decoding does not remove variance, and we make no such claims. Most papers and leaderboards do not report the confidence intervals, likely to avoid cluttered tables that do not convey sufficiently useful information. Fortunately, the confidence intervals are mostly a property of the evaluation task. For example, on SWE-bench verified, we measured that a 4% difference is significant for all 90+ results on the leaderboard.
>
> > C7: …outperforming DeepSeekCoder-33B and WizardCoder-15B …without GPT-4o’s SWE-bench Verified number…
>
> Please kindly note that we did not compare DeepSeekCoder or WizardCoder in the paper because they are not widely used for SWE-bench related tasks. Meanwhile, we did include GPT-4o's score in Table 1 for both SWE-agent (23.2%) and Agentless (38.8%), and SWE-RL is superior (41.0%).
>
> > C8: …no analysis of reward variance…
>
> Please kindly refer to Figure 5 in the paper. We show that the training reward steadily increases during training. It’s stable and the variance is low.
>
> > C9: The authors should pay special attention to… words and the choice of color
>
> We appreciate the suggestions. We will improve the clarity of the paper in the revision.
>
> > C10: What does the Python difflib do?
>
> As we explained in the paper, `difflib` computes sequence alignments to measure textual similarity; we use it (e.g., via `SequenceMatcher` on unified diffs) to score overlap between predicted and target patches, yielding a dense reward for localized edits.
>
> > C11: …lack of deeper reward design and validation…
>
> We acknowledge the reward is intentionally simple, but training was stable with low variance (C8) and we observe emergent, multi-step reasoning behaviors in traces (Q1/Q2). Combining SWE-RL reward is also better than using execution-only reward alone (Q1).
>
> > C12: …general reasoning improvement… remains unsubstantiated…
>
> We compare an SFT baseline trained on the same PR data (Q3) and also present behavioral diagnostics (Q2), both indicating that RL drives the out-of-domain gains while SFT leads to memorization. We will clarify this point in the revision.
>
> > C13: SWE-RL appears novel… without critical innovation.
>
> As we explained in Q1, the main novelty of SWE-RL lies in the insight that real-world software bugs, unlike open-ended code generation, often follow constrained, localized patterns, aligning with the plastic surgery hypothesis [1] in program repair. The insight itself does not seem novel as it has been known for a decade, but SWE-RL is the first to show that the simple difflib-based reward signal can already enable scalable RL on massive real-world software data. This finding will impact lots of future work in this critical application domain.
>
> > C14: …reference SWE papers from among these top-tier venues…
>
> Great suggestion. We’ll expand Related Work to cite representative papers from top-tier software engineering venues in the revision.
>
> [1] Barr et al. The Plastic Surgery Hypothesis
>
> [2] Pan et al. Training Software Engineering Agents and Verifiers with SWE-Gym
>
> [3] Yu et al. UTBoost: Rigorous Evaluation of Coding Agents on SWE-Bench
>
> [4] DeepSeek-AI. DeepSeek-R1: Incentivizing Reasoning Capability in LLMs via Reinforcement Learning

---

> > ### Comment · Reviewer_2vNV · 2025-08-05
> >
> > The authors have done a great job in responding to my comments, even though I am still skeptical on some key issues, for example:
> > - A patch with high textual overlap may still break invariants, violate specifications, or introduce subtle bugs.
> > - difflib does not seem to reason about control/data flow, side effects, or test coverage.
> > - I am not quite convinced that the learned model would not over-optimize on formatting or superficial token reuse (aligning with the plastic surgery hypothesis), while missing deeper, bug-fixing behavior.
> > - In software engineering, reproducibility and configuration stability are essential. RL methods are notoriously unstable, and without robustness checks, practitioners adopting this framework may fail to reproduce improvements.

---

> ### Author Response · Authors · 2025-08-05
>
> Thank you for taking the time to read our response and for your additional feedback. We truly appreciate it! We address your additional concerns as follows:
>
> > * A patch with high textual overlap may still break invariants, violate specifications, or introduce subtle bugs.
> > * difflib does not seem to reason about control/data flow, side effects, or test coverage.
> > * I am not quite convinced that the learned model would not over-optimize on formatting or superficial token reuse (aligning with the plastic surgery hypothesis), while missing deeper, bug-fixing behavior.
>
> We agree that the similarity reward is not perfect, but as we demonstrated in Q1, it can efficiently capture partial correctness of code changes and make large-scale training possible. Over the course of training, the model learns to produce semantically correct patches with this reward signal. In contrast, execution-based rewards require expensive environment setup and heavy data curation that limit scalability.
>
> To support this point, we provided a concrete example in Q1 showing that SWE-RL can trigger emergent reasoning patterns rather than superficial token reuse:
> ```
> We need to identify where the issue of not preserving…
> But wait, in the ToDoItem constructor, there's this line: _type = description…
> but that's not the issue…
> But then I saw it: in the second constructor…
> The actual issue is likely…
> And then it hits me…
> I think I've got it…
> ```
>
> Meanwhile, we conducted an additional experiment showing that combining the similarity reward with execution is superior to using execution-based reward alone:
>
> |Setting|Pass@1|
> |-|-|
> |Baseline|0.8%|
> |Execution-only|11.0%|
> |Execution + similarity reward|**14.2%**|
>
> These results indicated that the `difflib`-based similarity reward, despite its simplistic design, can essentially bootstrap the model's general reasoning capability and enhance its bug-fixing ability through reinforcement learning.
>
> > * In software engineering, reproducibility and configuration stability are essential. RL methods are notoriously unstable, and without robustness checks, practitioners adopting this framework may fail to reproduce improvements.
>
> We fully agree that reproducibility and stability are essential. To ensure reproducibility, we have described our training hyperparameters clearly in the paper. We’ll also open source the pipeline, model, data, and also all evaluation results after careful privacy reviews.
>
> As we indicated in C3, exhaustive ablations over all hyperparameter and component combinations are infeasible under our current compute budget; however, we will incorporate this valuable suggestion and add more ablations when we have sufficient computing resources. Despite these resource constraints, we conducted as many additional experiments as possible to show that our method is genuinely effective rather than noise. For example, the table below shows that SWE-RL consistently outperforms the SFT baseline across different prompt setups.
>
> |Benchmark|SFT (prompt in paper)|SFT (RL prompt)|RL (RL prompt)|
> |-|-|-|-|
> |HumanEval+|73.2|70.1|**79.9**|
> |MATH|71.7|70.6|**73.7**|
>
> Regarding stability, as we showed in Figure 5 of the paper, the reward steadily increases during the training with a low variance. In the table below, we also provide concrete reward scores in different training steps, showing the increasing trend:
>
> |Training Step|Reward|
> |-|-|
> |100|-0.84|
> |200|-0.38|
> |400|0.04|
> |800|0.17|
> |1600|0.22|
>
> Thank you again for your feedback. We hope our response could address your additional concerns. Should you have any new questions, please don't hesitate to let us know.

---

> > ### Comment · Reviewer_2vNV · 2025-08-08
> >
> > I sincerely thank the authors for their response to my concerns. I acknowledge that the computation budget could be a valid constraint. However, some theoretical framework could still be proposed, and the authors could point out the limitations for future research directions.

---

### Official Review · Reviewer_QFgA · 2025-06-28

**Clarity:** 3
**Significance:** 2
**Originality:** 3
**Rating:** 4
**Confidence:** 3

**Summary:**

The paper proposes SWE-RL to train LLMs on Github PR data with GRPO algorithm for software engineering tasks. Specifically, Agentless Mini, a revised version of Agentless is adopted as the scaffold for both rollout and evaluation. Reward is calculated as the sequence similarity between the generated and the golden patch. Results demonstrate that SWE-RL achieves SOTA performance among all models under 100B. It not only surpasses the corresponding SFT baseline, but also generalizes to other tasks beyond SWE-Bench.

**Questions:**

* It is unexpected to see degradation of SFT model on general tasks in Table 3 with respect to the base model, given that coding and general SFT data are also included in training. Do you have any insights?

**Ethical Concerns:**

["NO or VERY MINOR ethics concerns only"]

**Final Justification:**

The rebuttal mostly addressed my concerns on the evaluation part. The primary reason for not giving a higher score is that edit similarity as a reward in SWE tasks is now overshadowed by execution feedback which is more precise and widely adopted, limiting the practical impact of this work.

**Limitations:**

Yes.

**Quality:**

3

**Strengths And Weaknesses:**

Strengths:
* The proposed RL framework is novel in software engineering domain.
* The simple reward of sequence similarity is easily scalable and demonstrated effective.
* SWE-RL not only improves performance in the targeted domain but also generalizes to other tasks where reasoning is helpful.
* The paper is written clearly.

Weaknesses:
* Did you employ any test-time scaling during RL training by generating multiple patches and selecting only one for each rollout? If not, what's the reason for reporting the evaluation metrics with test-time scaling, rather than just let the model generate one patch and submit for evaluation? It can be more intuitive to have training and evaluation under the same setting.
* The comparison between the RL and the SFT model on the original SWE-Bench verified task (instead of the repair task) is critical but not included.
* Have you tried SFT with synthetic code editing data only? While this may degrade performance on general benchmarks, such an experiment can provide an upper bound of SFT performance under the assumption that the downstream task is fixed and known. It would be great to see SWE-RL is better even under this setting.

---

> ### Author Rebuttal · Authors · 2025-07-31
>
> Dear Reviewer QFgA, we deeply appreciate your insightful feedback and suggestions for our work. In our responses below, we address each primary question (denoted as Q) and comment (denoted as C). Additionally, we will revise our paper to incorporate editorial suggestions. Should there be any misunderstandings of the questions, please kindly let us know; we are eager to communicate with you throughout the discussion period.
>
>
> > Q1: It is unexpected to see degradation of SFT model on general tasks in Table 3 with respect to the base model, given that coding and general SFT data are also included in training. Do you have any insights?
>
> Great question. We believe that the underlying reason is that RL teaches the model generalized reasoning while SFT only lets the model memorize the data it is trained on, as supported by a recent paper [1]. SWE-RL, while being a single RL task, requires the model to do complex reasoning on the long context input comprising hundreds to thousands of lines of code to correctly localize the bug and generate the patch. This process incentivizes the model to learn generalized reasoning.
>
> > C1: Did you employ any test-time scaling during RL training...
>
> Thank you for the thoughtful question. No, we did not use test‑time scaling during RL training: each rollout produces a single patch and is rewarded via patch similarity, and we do not train any separate selector or reranker. We still report test‑time–scaled evaluation to follow existing work like Agentless [2], also validating the generalizability of SWE-RL where the reasoning learned through SWE-RL transfers to other subtasks. As a result, in training, we only need to guide the model to generate the best patch for each rollout, keeping the RL scalable and execution‑free. For a matched setting, our pass@1 (no scaling) results are reported in Table 2 and show consistent gains over SFT.
>
>
> > C2: The comparison between the RL and the SFT model on the original SWE-Bench verified task (instead of the repair task) is critical but not included.
>
> Great point. The main reason why we evaluate the SFT and RL model on the repair task is to focus on comparing their bug fixing and code editing capabilities, isolating the effect of localization used in the Agentless framework. To have a better understanding of their respective performance on the full SWE-bench Verified task, we conducted an additional experiment below to show the SFT pass@1 with the end to end pipeline. From the table, we can see that SFT is worse than RL by 4.8 points. According to our statistical analysis, a 4-point difference is statistically significant enough to show that RL is better than SFT on SWE-bench Verified.
>
> |Setting|Pass@1|
> |-|-|
> |SFT|36.2%|
> |RL|**41.0%**|
>
>
> > C3: Have you tried SFT with synthetic code editing data only?...
>
> Good question. We mixed different SFT data to make sure the model can follow different kinds of instructions and can successfully complete all the steps in Agentless-Mini. In the table below, we also conducted an SFT experiment with only code editing data. It has a similar score to the mixed SFT model in the repair only setting. This checkpoint cannot be evaluated in the other Agentless steps because it cannot follow instructions well and suffers from many format errors in the other subtasks such as localization and test generation.
>
> |Setting|Repair Pass@1|
> |-|-|
> |SFT|29.6%|
> |SFT (editing only)|29.8%|
> |RL|**34.8%**|
>
>
> [1] Chu et al. SFT Memorizes, RL Generalizes: A Comparative Study of Foundation Model Post-training.
>
> [2] Xia et al. Agentless: Demystifying LLM-Based Software Engineering Agents

---

> > ### Comment · Reviewer_QFgA · 2025-08-03
> >
> > Thank you for the response and particularly the additional results. I now get the point of using the repair task to specifically evaluate what your RL aims to teach the model. It is still somewhat surprising to me that the editing-only SFT brings almost no gain upon general SFT, probably because the SFT data is not as good as the RL one for SWE-Bench tasks. But overall I believe the evaluation is solid and establishes the effectiveness of the method.

---

> > > ### Author Response · Authors · 2025-08-03
> > >
> > > Thank you for taking the time to read our response. We truly appreciate it!
> > >
> > > > It is still somewhat surprising to me that the editing-only SFT brings almost no gain upon general SFT, probably because the SFT data is not as good as the RL one for SWE-Bench tasks.
> > >
> > > Great point! We want to kindly clarify that the same code-editing SFT dataset is used consistently in both editing-only and general SFT runs. Our view is that, in general SFT, mixing editing-only with general coding and dialogue enables the model to learn broad skills (e.g., code reasoning and instruction following), some of which are transferable to SWE-Bench. Hence, the editing-only SFT adds little headroom. In contrast, the RL stage, guided by the SWE-RL reward, bootstraps the model's generalized reasoning skills, leading to better performance.
> > >
> > > We hope this explanation is helpful. Should you have any new questions or concerns, please don't hesitate to let us know.

---

### Official Review · Reviewer_ZGKJ · 2025-06-30

**Clarity:** 3
**Significance:** 2
**Originality:** 3
**Rating:** 4
**Confidence:** 4

**Summary:**

This paper proposes SWE-RL, a reinforcement learning framework to fine-tune Large Language Models (LLMs) on GitHub issue–patch data. The authors train a 70B Llama3-based model using a character-level difflib reward and the GRPO algorithm, achieving 41.0% on the SWE-bench Verified benchmark—a new state-of-the-art for open-source models of this size. The authors further claim that this training method improves the model’s general reasoning ability, as evidenced by improvements on out-of-domain tasks including MATH, HumanEval+, and MMLU.

**Questions:**

1. Your central claim is that reinforcement learning on software repair tasks leads to improved general reasoning. However, this remains an observational finding. Could you clarify what specific mechanism or training dynamics enable skill transfer from patch generation to tasks like MATH or HumanEval+?

2. The RL-trained model is evaluated using a detailed chain-of-thought (CoT) prompt that could itself induce improved behavior. To attribute improvements to RL rather than prompting, did you evaluate the SFT baseline using the exact same prompt as used during RL training and evaluation? If not, how can you isolate the effect of RL from the prompt’s contribution?


3. The difflib-based reward function operates on surface-level textual similarity and is agnostic to semantic correctness. Have you encountered examples where high-reward patches are semantically incorrect or logically invalid? How do such cases affect the interpretation of “improved reasoning”?

4. Your RL formulation treats patch generation as a single-step decoding task, without intermediate actions or sequential planning. How does this setting support the emergence of non-trivial reasoning strategies, as opposed to shallow pattern recognition?

5. The observed gains in task accuracy are not sufficient to support the claim that RL improves reasoning. To substantiate this, please provide concrete behavioral evidence that the model acquired new reasoning capabilities. such as generating longer or more structured solutions, using intermediate steps more effectively, or demonstrating improved error correction. Alternatively, comparative analysis of model behavior (e.g., reasoning trace complexity or error type distribution) before and after RL would strengthen the claim.

If the authors can provide more details to address my comments, I would be inclined to raise my score.

**Ethical Concerns:**

["NO or VERY MINOR ethics concerns only"]

**Final Justification:**

The following points summarize my reasoning:
1. **Generalization Evidence**   The authors provide concrete behavioral evidence (e.g., longer reasoning traces, increased output length) and controlled prompt experiments that support the claim that RL training improves general reasoning ability.

2. **Reward Design Justification**   Although the difflib-based reward is surface-level, the authors justify its scalability and show that it complements execution-based rewards, enabling stable and large-scale RL training.

3. **Empirical Strength**   The performance on SWE-bench Verified and out-of-domain tasks is competitive, with consistent improvements over SFT baselines trained on the same data.

4. **Remaining Limitations**   The work lacks deeper theoretical explanation and more thorough ablations (e.g., KL penalty, group size), but these are not deal-breakers given the strong empirical results.

Overall, I think the contribution is shareable with the community.

**Limitations:**

Yes

**Paper Formatting Concerns:**

There are no major formatting issues observed in this paper. The paper is well-formatted and adheres to the required NeurIPS standards.

**Quality:**

3

**Strengths And Weaknesses:**

## Strengths

1. The paper presents a strong empirical result, achieving a 41.0% solve rate on SWE-bench Verified, which demonstrates that reinforcement learning can be effectively applied to large-scale, realistic software data.

2. The experimental setup is clean and consistent—both the SFT and RL models are trained on the same dataset with aligned prompt templates, allowing for a meaningful attribution of performance gains to the training method itself.

3. The paper is well-written, clearly organized, and easy to follow. The structure facilitates understanding of both the implementation details and the high-level motivations.

4. The observed improvements on out-of-domain tasks like mathematics and general code reasoning, though small, are intriguing. These results could inspire future research into how specialized training in one domain affects generalization capabilities in others.

## Weaknesses

1. The paper makes limited methodological contributions. The problem formulation follows existing benchmarks like SWE-bench, and the approach is a direct combination of known components: GRPO for optimization, a difflib-based reward signal, and chain-of-thought prompting. There is no new algorithmic mechanism, inductive bias, or novel architecture introduced. As such, the contribution lies more in implementation scale than in conceptual innovation.

2. The central claim that RL improves general reasoning ability is weakly supported. The reward function is based purely on token-level similarity, which does not capture semantic correctness or reasoning structure. The authors do not provide a theoretical or causal argument explaining why optimizing patch similarity should lead to improvements on tasks like math or logical reasoning, nor do they offer any behavioral or process-level evidence to validate such a connection.

3. The RL formulation is shallow relative to the complexity of the reasoning tasks it claims to improve. The model operates in a single-step generation setting, with no planning, intermediate decision-making, or multi-turn interaction. The reward signal cannot distinguish between semantically different outputs if they are syntactically similar, further limiting the model’s capacity to learn deeper problem-solving strategies. This makes it likely that the model is learning format-aware mimicry rather than genuine reasoning.

4. The evaluation design does not adequately isolate the effects of RL. While the SFT and RL models are trained on the same dataset, they are evaluated under different prompting conditions, and there is no ablation study to determine how much of the observed performance gain stems from RL versus prompting. Without a controlled comparison using the same prompt template, the attribution of improvement to RL is not justified.

5. The reported cross-task improvements are small and not convincingly explained. Although the paper highlights slight gains on unrelated tasks such as mathematics, it does not analyze whether these improvements result from behavioral changes in the model or merely reflect incidental correlations. No trajectory analysis, failure case study, or reasoning trace inspection is provided to strengthen the generalization claim.

---

> ### Author Rebuttal · Authors · 2025-07-31
>
> Dear Reviewer ZGKJ, we deeply appreciate your insightful feedback and suggestions for our work. In our responses below, we address each primary question (denoted as Q) and comment (denoted as C). Should there be any misunderstandings of the questions, please kindly let us know; we are eager to communicate with you throughout the discussion period.
>
> > Q1: …what specific mechanism or training dynamics enable skill transfer from patch generation to tasks like MATH or HumanEval+?
>
> Great question. In SWE-RL training, the model needs to generate a correct patch in one step conditioned on the relevant code context that is typically long. To maximize the reward, it needs to perform extensive and accurate reasoning to find the correct edit location and to produce a correct patch. This process involves significant reasoning and helps the model bootstrap its general reasoning capabilities that are transferable to other tasks. In the table below, we can observe that the maximum thinking length (measured in character count) continuously increases during training:
>
> |Training steps|Max thinking length|
> |-|-|
> |400|**15546**|
> |800|**17704**|
> |1600|**20203**|
>
> Similarly, in the following table, we measured the average output length of the original, SFT, and SWE-RL checkpoints on HumanEval+ and MATH. After RL, the reasoning is longer, supporting our assumption that the model acquires general reasoning ability through improved software issue solving.
>
> |Benchmark \ Setting|Baseline|SFT|RL|
> |-|-|-|-|
> |HumanEval+|746|657|**1622**|
> |MATH|1876|1643|**2552**|
>
> > Q2: …did you evaluate the SFT baseline using the exact same prompt as used during RL training and evaluation?…
>
> We use the most appropriate prompts tailored to each model checkpoint during evaluation. For SWE-bench, both the SFT and RL checkpoints are evaluated using identical system and user prompts. For out-of-domain benchmarks like HumanEval+ and MATH, we keep the user prompts the same across checkpoints, but apply the system prompt (shown in Figure 2) only to the RL-trained model. This is because the SFT checkpoint's training data includes system prompts for code editing tasks, but not for general coding tasks. As a result, applying a system prompt in those general tasks would be out-of-distribution for the SFT model. In contrast, the RL checkpoint is expected to generalize its reasoning under the system prompt, making its use appropriate.
>
> To confirm that this difference in prompting doesn't unfairly benefit the RL model, we ran an additional experiment applying the system prompt to the SFT checkpoint on HumanEval+ and MATH as well. This allowed us to test whether the system prompt itself influences performance on general tasks for SFT.
>
> |Benchmark|SFT (prompt in paper)|SFT (RL prompt)|RL (RL prompt)|
> |-|-|-|-|
> |HumanEval+|73.2|70.1|**79.9**|
> |MATH|71.7|70.6|**73.7**|
>
> From the result, we can see that the SFT checkpoint is still worse than the RL one despite the prompt alignment. It’s also worse than the prompt we used for evaluation in the paper. It matches our claim that RL leads to generalized reasoning while SFT is more about memorizing the data patterns.
>
> > Q3: The difflib-based reward function operates on surface-level textual similarity and is agnostic to semantic correctness…
>
> We want to kindly highlight that the main novelty of SWE-RL lies in the insight that real-world software bugs, unlike open-ended code generation, often follow constrained, localized patterns, aligning with the plastic surgery hypothesis [1] in program repair. The difflib-based reward can efficiently capture partial correctness of code changes.
> While we do observe some high-reward patches that fail the tests, these episodes can still improve reasoning because the difflib signal implicitly incentivizes accurate bug localization and minimal, targeted edits, which are necessary intermediate steps toward a correct fix. With more training steps, the model can learn to refine these patches further and produce semantically correct solutions.
> Crucially, SWE-RL reward is what makes large-scale training possible; execution-based rewards require expensive environment setup and heavy data curation, limiting scalability. For example, SWE-Gym [2] requires 200 human annotation hours and 10,000 CPU core hours to produce 2438 trainable instances, while we collect 273k instances for training fully automatically.
>
> In the meantime, test-based reward is never perfect. In certain cases, it can be worse than similarity metrics where incorrect patches pass all the tests but miss the true intention of the problem. This claim is well supported by the literature. For example, [3] shows that the insufficient tests in SWE-bench impacts 40.9% of SWE-Bench Lite and 24.4% of SWE-Bench Verified leaderboard entries. Below is an example illustrated in the paper, where the original issue requires a function that computes polynomial fits for data, to handle missing data in the inputs x and y. However, the original test case only considers scenarios where both x and y have missing data. The generated patch, while passing the test, fails to handle additional cases where only one input has missing data.
>
> ```
> # Insufficient test
> def test_missing_data(self, df):
>   groupby = GroupBy(["group"])
>   df.iloc[5:10] = np.nan
>   res1 = PolyFit()(df[["x", "y"]], groupby, "x", {})
>   res2 = PolyFit()(df[["x", "y" ]].dropna(), groupby , "x", {})
>   assert_frame_equal(res1, res2)
>
> # Incorrect patch that passes the test
> def _fit_predict(self, data):
>   y = data["y"].dropna()
>   x = data["x"].dropna()
>   if x.shape[0] != y.shape[0]:
>     raise ValueError ("x and y must have the same number of non - missing values ")
>   if x.nunique() <= self.order:
>     # TODO warn ?
>     xx = yy = []
> ```
>
> To further support our point, we did an additional experiment showing that combining SWE-RL reward with execution is superior to applying execution reward alone. The results are in the table below. Please kindly refer to Q1 from Reviewer n4Dr for more details:
>
> |Setting|Pass@1|
> |-|-|
> |Baseline|0.8%|
> |Execution-only|11.0%|
> |Execution + SWE-RL reward|**14.2%**|
>
> > Q4: …How does this setting support the emergence of non-trivial reasoning strategies, as opposed to shallow pattern recognition?
>
> As we explain in Q3, SWE-RL requires non-trivial reasoning to localize errors and generate patches correctly to solve the software issues. During RL, such reasoning patterns emerge, involving planning and backtracking. We show one example below. Please also kindly refer to Figure 3 in the paper for more examples.
>
> ```
> We need to identify where the issue of not preserving single quotes in TODO comments occurs… In the ToDoItem class, the description is being extracted…
> But wait, in the ToDoItem constructor, there's this line: _type = description.Substring(0, idxDelimiter);. However, the idxDelimiter is being found …but that's not the issue here.
> But then I saw it: in the second constructor of ToDoItem…
> The actual issue is likely due to… And then it hits me… I think I've got it now.
> ```
>
> > Q5: …please provide concrete behavioral evidence that the model acquired new reasoning capabilities…
>
> Great point. As we showed in Q1 and Q2, the reasoning length of the model increases after SWE-RL in both training and evaluation. According to Figure 3 in the paper and the example we showed in Q4, SWE-RL triggers the model’s emergent reasoning skills such as planning, self-reflection, and divide-and-conquer. Below, we also show a concrete example on MATH where the model doesn’t do much reasoning before SWE-RL, but reasons deeply afterward:
>
> A math problem whose ground-truth answer is $\frac{\sqrt{21}}{5}$:
>
> Before SWE-RL
> ```
> We have that $\sin R = \frac{2}{5} = \frac{RS}{RT}$, so $RS = 2$ and $RT = 5$.  By the Pythagorean theorem… Then $\sin T = \frac{RS}{RT} = \frac{2}{5}$.
>
> Answer: $\frac{2}{5}$
> ```
>
> After SWE-RL:
> ```
> To find $\sin{T}$, we first need to understand the given information. We're dealing with a right-angled triangle $\triangle{RST}$, and it's given that $\sin{R}=\frac{2}{5}$…
> From the diagram, we see that the length of the hypotenuse…
> Next, to find $\sin{T}$, we need to determine the length of the side opposite…
> However…
> Given $RS = 2$ and $RT = 5$…
> But, we actually need to use $ST$ and $RT$ …
> Thus, $\sin{T} = \frac{ST}{RT} = \frac{\sqrt{21}}{5}$.
> </think>
> <solution>
> Answer: $\frac{\sqrt{21}}{5}$
> </solution>
> ```
>
> > C1: The paper makes limited methodological contributions…
>
> As we explained in Q3, the main novelty of SWE-RL lies in the insight that real-world software bugs, unlike open-ended code generation, often follow constrained, localized patterns, aligning with the plastic surgery hypothesis [1] in program repair. Furthermore, SWE-RL is the first to show that the simple difflib-based reward signal can already enable scalable and effective RL on massive real-world software data. This finding will impact lots of future work in this critical application domain.
>
> > C2: The reported cross-task improvements are small and not convincingly explained…
>
> Indeed, 1-2% performance gains on HumanEval or CRUXEval are not likely to be significant by themselves (at the 0.05 level). That is why we reported on multiple evaluations where all results are consistently in favor of RL, thus giving increased significance (for example, via Fisher’s combined probability test). To support this, we conducted additional statistical testing (link omitted due to rebuttal policy), showing that >0.8% on MMLU (14k examples), 3% on CRUXEval, and >3% on the full MATH are already significant. So the combination of our results will reach significance at the 0.05 level. We will include this statistical analysis in the revision of our paper.
>
> > C3/C4/C5 are covered in the previous responses.
>
> [1] Barr et al. The Plastic Surgery Hypothesis.
>
> [2] Pan et al. Training Software Engineering Agents and Verifiers with SWE-Gym.
>
> [3] Yu et al. UTBoost: Rigorous Evaluation of Coding Agents on SWE-Bench.

---

> > ### Comment · Reviewer_ZGKJ · 2025-08-03
> >
> > Thank you for the detailed and thoughtful rebuttal. I appreciate the additional experiments on prompt alignment, reward signal combination, and reasoning behavior analysis. The evidence showing longer and more structured outputs after RL training addresses my concerns regarding generalization. While the difflib-based reward remains surface-level, the authors make a convincing case for its scalability and effectiveness when combined with execution-based feedback. Although the work would benefit from deeper theoretical insights and more comprehensive ablations, overall, the empirical contribution is solid.

---

> > > ### Author Response · Authors · 2025-08-03
> > >
> > > Thank you for taking the time to read our response. We truly appreciate it! Should you have any new questions or concerns, please don't hesitate to let us know.

---

### Official Review · Reviewer_n4Dr · 2025-07-03

**Clarity:** 3
**Significance:** 2
**Originality:** 3
**Rating:** 4
**Confidence:** 4

**Summary:**

This paper proposes SWE-RL, a novel reinforcement learning (RL) approach that trains large language models (LLMs) to solve real-world software engineering (SE) tasks using open-source software evolution data, i.e., GitHub pull requests (PRs). Unlike previous RL methods that focused on competitive programming or math, SWE-RL targets real-world issue resolution, leveraging rule-based rewards derived from patch similarity between model-generated and human-written code. The authors train Llama3-SWE-RL-70B using this approach and evaluate it on SWE-bench Verified, where it achieves a 41.0% solve rate, surpassing all open-source LLMs under 100B parameters and rivaling proprietary models like GPT-4o. Surprisingly, this RL training also improves general reasoning ability across out-of-domain tasks (e.g., math, MMLU), even outperforming supervised fine-tuning (SFT) baselines trained on more diverse datasets

**Questions:**

(1) Have the authors considered semantic diffing or static analysis tools (e.g., AST comparison or test outcome delta) for more robust rewards?

(2) Could test failures or runtime logs be incorporated as feedback in future RL training?

**Ethical Concerns:**

["NO or VERY MINOR ethics concerns only"]

**Limitations:**

Yes.

**Quality:**

3

**Strengths And Weaknesses:**

Strengths:

(1) Novel direction for RL training: SWE-RL extends RL training to software evolution data—a previously untapped but richly structured resource. It moves beyond execution-based or synthetic RL setups.

(2) Well-motivated reward function: The reward uses difflib-based patch similarity, allowing for partial credit and encouraging incremental improvements, which better reflects real-world issue resolution.

(3) Good performance and comprehensive experiments: Achieves the best performance among medium-sized open models (41.0%), without using proprietary model outputs for supervision. The paper conducts comprehensive experiments including rigorous baselines (SFT and original model), scaling studies, reward ablation, and cross-domain generalizability testing.

Weaknesses:

(1) Reward function limitation: As the author said in "Limitations", the reward is based on string-level patch similarity, not semantic equivalence. This penalizes functionally correct but syntactically different fixes, potentially limiting solution diversity.

(2) Limited interactivity: The pipeline is not interactive or agentic—it lacks tool use, exploration, or test-driven feedback during training. This may limit the model’s ability to generalize to more autonomous agent settings.

(3) Single-task RL training: The model is only RL-trained on one task (issue resolution). While generalization is observed, multi-task RL training might further enhance its capabilities.

(4) The training cost is too high, which requires 512 H100 GPUs for ~ 32 wall-time hours, which may limit accessibility and reproducibility despite using open-source models.

---

> ### Author Rebuttal · Authors · 2025-07-31
>
> Dear Reviewer n4Dr, we deeply appreciate your insightful feedback and suggestions for our work. In our responses below, we address each primary question (denoted as Q) and comment (denoted as C). Additionally, we will revise our paper to incorporate editorial suggestions. Should there be any misunderstandings of the questions, please kindly let us know; we are eager to communicate with you throughout the discussion period.
>
> > Q1: Have the authors considered semantic diffing or static analysis tools (e.g., AST comparison or test outcome delta) for more robust rewards?
>
> Great point. We applied basic patch normalization (e.g., header and whitespace removal) before comparison, but not at the AST level. Regarding test outcomes, we conducted an additional experiment incorporating unit-test results into reward calculation. As detailed in Q2, we also enabled the model to interact with the environment using tools such as Bash and an editor. Due to time constraints, this experiment was performed with Llama-3.1-8B-Instruct. The table below shows pass@1 on SWE-bench Verified comparing three settings: (1) the baseline Llama-3.1-8B-Instruct, (2) RL with a binary execution-only reward (fail = 0, pass = 1), and (3) RL with execution reward combined with the SWE-RL reward. All models were trained for 400 RL steps on the SWE-Gym [1] dataset.
>
> |Setting|Pass@1|
> |-|-|
> |Baseline|0.8%|
> |Execution-only|11.0%|
> |Execution + SWE-RL reward|**14.2%**|
>
> Both reward signals significantly improve over the baseline, and the SWE-RL reward complements execution-based rewards, aligning with our insight that software issues often require contextualized fixes, consistent with the Plastic Surgery Hypothesis [2] in program repair. While execution-based methods require substantial data collection in executable environments, the SWE-RL reward scales to large, open software corpora.
>
> > Q2: Could test failures or runtime logs be incorporated as feedback in future RL training?
>
> Absolutely! In the Q1 experiments, we allowed the model to issue Bash commands freely when solving issues. Although not strictly enforced, the model could execute tests and use them as feedback to improve its solutions. In the table below, we divide trajectories generated during RL by whether they observed and fixed test errors, and we report their pass@1.
>
> |Trajectory Pattern|Pass@1|
> |-|-|
> |No error fixing|22.4%|
> |Fixing test errors|**25.8%**|
>
> A concrete example of the LLM leveraging runtime logs for better issue solving:
>
> ```
> ...
> [assistant]
> Now that we've updated the import statement, we should be able to run the test again.
>
> Let's try running the test again to see if the import issue is resolved.
>
> <tool: bash>
> python -m unittest /testbed/tests/test_cognitoidp_models.py
> </tool>
> </bash>
>
> [user]
> Traceback (most recent call last):
>   File "<frozen runpy>", line 198, in _run_module_as_main
>   File "<frozen runpy>", line 88, in _run_code
>   File "/opt/miniconda3/envs/testbed/lib/python3.12/unittest/__main__.py", line 18, in <module>
>     main(module=None)
>   File "/opt/miniconda3/envs/testbed/lib/python3.12/unittest/main.py", line 104, in __init__
>     self.parseArgs(argv)
>   File "/opt/miniconda3/envs/testbed/lib/python3.12/unittest/main.py", line 153, in parseArgs
>     self.createTests()
>   File "/opt/miniconda3/envs/testbed/lib/python3.12/unittest/main.py", line 164, in createTests
>     self.test = self.testLoader.loadTestsFromNames(self.testNames,
>                 ^^^^^^^^^^^^^^^^^^^^^^^^^^^^^^^^^^^^^^^^^^^^^^^^^^
>   File "/opt/miniconda3/envs/testbed/lib/python3.12/unittest/loader.py", line 207, in loadTestsFromNames
>     suites = [self.loadTestsFromName(name, module) for name in names]
>               ^^^^^^^^^^^^^^^^^^^^^^^^^^^^^^^^^^^^
>   File "/opt/miniconda3/envs/testbed/lib/python3.12/unittest/loader.py", line 137, in loadTestsFromName
>     module = __import__(module_name)
>              ^^^^^^^^^^^^^^^^^^^^^^^
>   File "/testbed/tests/test_cognitoidp_models.py", line 5, in <module>
>     class TestCognitoResourceServer(unittest.TestCase):
>   File "/testbed/tests/test_cognitoidp_models.py", line 6, in TestCognitoResourceServer
>     @mock_cognito_idp
>      ^^^^^^^^^^^^^^^^
> NameError: name 'mock_cognito_idp' is not defined. Did you mean: 'mock_cognitoidp'?
>
> [assistant]
> The error message indicates that `mock_cognito_idp` is not defined, even though we've already updated the import statement to `mock_cognitoidp`.
>
> It looks like there's another issue with the usage of `mock_cognito_idp` in the `TestCognitoResourceServer` class. We should actually use `@mock_cognitoidp` instead of `@mock_cognito_idp`.
>
> Let's update the test file to use `@mock_cognitoidp` instead.
>
> <tool: edit>
> /testbed/tests/test_cognitoidp_models.py
> <<<<<<< SEARCH
>     @mock_cognito_idp
> =======
>     @mock_cognitoidp
> >>>>>>> REPLACE
> </tool>
>
> [user]
> File /testbed/tests/test_cognitoidp_models.py has been successfully updated at line 6 and now contains 22 lines.<|eot_id|>
>
> [assistant]
> Now that we've updated the decorator to use `mock_cognitoidp` correctly, we should be able to run the test again.
>
> Let's try running the test again to see if the usage issue is resolved.
>
> <tool: bash>
> python -m unittest /testbed/tests/test_cognitoidp_models.py
> </tool>
> </bash>
>
> [user]
> .
> ----------------------------------------------------------------------
> Ran 1 test in 0.038s
>
> OK
>
> ...
> ```
>
> These results indicate that incorporating tests improves issue solving, and we will include this valuable idea in future iterations.
>
> > C1: Reward function limitation: ...the reward is based on string-level patch similarity, not semantic equivalence. This penalizes functionally correct but syntactically different fixes, potentially limiting solution diversity
>
> We want to kindly highlight that the main novelty of SWE-RL lies in the insight that real-world software bugs, unlike open-ended code generation, often follows constrained, localized patterns, aligning with the plastic surgery hypothesis [2] in program repair. The difflib-based reward can efficiently capture partial correctness of code changes. Crucially, this reward is what makes large-scale training possible; execution-based rewards require expensive environment setup and heavy data curation [1], limiting scalability. SWE-RL unlocks scalable, generalizable RL on real-world software data for the first time.
>
> In the meantime, test-based reward is never perfect. In certain cases, it can be worse than similarity metrics where incorrect patches pass all the tests but miss the true intention of the problem. This claim is well supported by the literature. For example, EvalPlus [4] shows that the insufficient tests in the original HumanEval and MBPP benchmarks lead to many erroneous code solutions being accepted. Furthermore, [5] shows that the insufficient tests in SWE-bench impact 40.9% of SWE-Bench Lite and 24.4% of SWE-Bench Verified submissions. Below is an example illustrated in the paper, where the original issue requires a function that computes polynomial fits for data, to handle missing data in the inputs x and y. However, the original test case only considers scenarios where both x and y have missing data. The generated patch, while passing the test, fails to handle additional cases where only one input has missing data.
>
> ```
> # Insufficient test
> def test_missing_data(self, df):
>   groupby = GroupBy(["group"])
>   df.iloc[5:10] = np.nan
>   res1 = PolyFit()(df[["x", "y"]], groupby, "x", {})
>   res2 = PolyFit()(df[["x", "y" ]].dropna(), groupby , "x", {})
>   assert_frame_equal(res1, res2)
>
> # Incorrect patch that passes the test
> def _fit_predict(self, data):
>   y = data["y"].dropna()
>   x = data["x"].dropna()
>   if x.shape[0] != y.shape[0]:
>     raise ValueError ("x and y must have the same number of non - missing values ")
>   if x.nunique() <= self.order:
>     # TODO warn ?
>     xx = yy = []
> ```
>
> Our insights are also backed by the results we showed in Q1 and Q2, where combining SWE-RL reward can surpass execution-only signal in a multi-turn RL setup.
>
> > C2: Limited interactivity: The pipeline is not interactive or agentic—it lacks tool use, exploration, or test-driven feedback during training...
>
> Thank you for the thoughtful point. Our design choice was to prioritize scalability: SWE‑RL trains on large, real‑world repos without execution environments. The reward is orthogonal to interactivity; as Q1–Q2 show, adding it to agentic, tool-using RL improves over execution‑only signals. We view the practical recipe as hybrid, first training at scale with SWE‑RL, then doing agentic RL on executable subsets with test‑driven feedback. We’ll clarify this trade‑off and our roadmap in the paper.
>
> > C3: Single-task RL training: ...While generalization is observed, multi-task RL training might further enhance its capabilities.
>
> Great suggestion! We plan to explore multi-task RL training in future work to further understand the capability transfer and generalization.
>
> > C4: The training cost is too high, which requires 512 H100 GPUs for ~ 32 wall-time hours, which may limit accessibility and reproducibility despite using open-source models.
>
> We acknowledge the relatively high training cost. However, this cost is expected for doing RL at scale on software engineering tasks and remains affordable for industry labs (e.g., agentic systems like Kimi K2 [3] operate with over 10k parallel containers during training, substantially larger GPU and CPU cost). To ensure reproducibility, we have described our training hyperparameters clearly in the paper. We’ll also open source the pipeline, model, data, and also all evaluation results after careful privacy reviews.
>
> [1] Pan et al. Training Software Engineering Agents and Verifiers with SWE-Gym.
>
> [2] Barr et al. The Plastic Surgery Hypothesis.
>
> [3] Kimi Team. Kimi K2: Open Agentic Intelligence.
>
> [4] Liu et al. Is Your Code Generated by ChatGPT Really Correct? Rigorous Evaluation of Large Language Models for Code Generation.
>
> [5] Yu et al. UTBoost: Rigorous Evaluation of Coding Agents on SWE-Bench.

---

### Note · Authors · 2025-08-11

Dear Reviewers, ACs, and SACs,

We deeply appreciate your thoughtful reviews, active discussions, and continued support of our work. We are encouraged by the recognition of SWE-RL as a novel and scalable RL framework leveraging open software data (Reviewers n4Dr, ZGKJ, QFgA, and 2vNV), and value the acknowledgment of our strong empirical results and clean SFT-vs-RL setup (Reviewer ZGKJ), clear writing (Reviewers ZGKJ and QFgA), comprehensive and solid evaluation (Reviewer n4Dr and QFgA), and the emphasis on semantics and stability (Reviewer 2vNV).

We also appreciate the questions raised by each reviewer, and believe that we have addressed most key concerns. Below are the most important points.

> Reward is surface-level (difflib), interactivity and semantics not captured

In a controlled 8B setting, we show that combining SWE-RL reward in an interactive multi-turn setup is better than applying execution-only reward, meaning that the similarity reward is not surface-level, but incentivizes the model to produce semantically correct patches over the RL training.

|Setting|Pass@1|
|-|-|
|Baseline|0.8|
|Execution-only|11.0|
|Execution + SWE-RL reward|**14.2**|

We also provide additional behavioral evidence and empirical examples showing that SWE-RL reward encourages the model to think longer and discover new reasoning patterns.

|Training steps|Max thinking length|
|-|-|
|400|15546|
|800|17704|
|1600|20203|
```
We need to identify where the issue…
But wait, in the ToDoItem constructor…
but that's not the issue…
But then I saw it: in the second…
The actual issue is…
And then it hits me…
I think I've got it…
```
> Gains outside SWE may be due to prompting

We used the most appropriate prompts in the paper. With prompt alignment, SFT remains below RL, so improvements persist beyond prompting.
|Benchmark|SFT (prompt in paper)|SFT (RL prompt)|RL (RL prompt)|
|-|-|-|-|
|HumanEval+|73.2|70.1|**79.9**|
|MATH|71.7|70.6|**73.7**|

> End-to-end comparison on SWE-bench Verified

In the full pipeline: RL is better than SFT by 4.8 points, exceeding the ~4-point 0.05 significance threshold we measured for the benchmark.
|Setting|Pass@1|
|-|-|
|SFT|36.2|
|RL|**41.0**|

> Does RL beat SFT trained on editing-only data?

RL is better than both mixed SFT and SFT trained on editing-only data.

|Setting|Repair Pass@1|
|-|-|
|SFT|29.6|
|SFT (editing only)|29.8|
|RL|**34.8**|

We hope these results provide clear, decision-relevant evidence in favor of our submission.

Best,

Authors

---

### Decision · Program_Chairs · 2025-09-17

**Decision:**

Accept (poster)

**Comment:**

This paper proposes SWE-RL, a novel reinforcement learning framework that fine-tunes large language models (LLMs) on real-world software evolution data from GitHub pull requests. The core scientific claim is that this training method, which uses a simple difflib-based patch similarity as a reward signal, not only achieves state-of-the-art performance for open-source models on the SWE-bench Verified benchmark (41.0% solve rate) but also enhances the model's general reasoning abilities, leading to improved performance on out-of-domain tasks like MATH and HumanEval+. The paper's main strengths are its novel application of RL to this domain at scale, the scalability and effectiveness of its reward function, and its comprehensive experimental evaluation. The primary weaknesses noted by reviewers were the shallow nature of the difflib reward, which lacks semantic understanding, and the initial lack of convincing evidence for the generalization claim. The high computational cost and missing ablations were also cited as limitations.

The authors' response addressed some of the reviewers' concerns. The authors conducted several new experiments. They demonstrated that combining the SWE-RL reward with execution-based rewards is superior to using execution-only rewards, justifying the use of their simple signal. They also ran a controlled experiment to show that the RL model's performance gains are not due to prompting but are a result of the RL training itself. To support their generalization claim, the authors provided quantitative evidence of increased "thinking length" and output length after RL training, along with qualitative examples of new reasoning patterns. Finally, they provided a direct end-to-end comparison, showing that the RL model significantly outperforms the SFT baseline on SWE-bench Verified. As a result of the discussion, all reviewers, including the highly confident reviewer 2vNV, updated their final justifications to be more positive, acknowledging that their concerns were "meaningfully improved" and their confidence in the results was "adequately improved." In my final decision, I weighed the authors' successful efforts to address the most critical empirical and validity concerns very highly, while recognizing that some remaining limitations—such as the lack of full hyperparameter ablations—are acceptable given the strong empirical results and the high computational cost.